# UNIFIED DATA SELECTION FOR LLM REASONING

## ABSTRACT

Effectively training LLMs for complex, long-CoT reasoning is often bottlenecked by the need for massive high-quality reasoning data. Existing methods are either computationally expensive or fail to reliably distinguish high- from low-quality reasoning samples. To address this, we propose High-Entropy Sum (HES)—a training-free metric that sums only the entropy of the top 0.5% highest-entropy tokens in each reasoning sample, focusing on critical forking points to better capture reasoning quality. We validate HES across three mainstream training paradigms: SFT, RFT, and RL. In SFT, training on just the top 20% of data ranked by HES matches full-dataset performance, while using the lowest-HES data severely degrades it. In RFT, HES-based selection outperforms random baseline. In RL, pairing highest-HES successful trajectories with random failed ones enables the model to learn both strong reasoning patterns and diverse failure modes, significantly surpassing existing training-free selection methods. Our findings establish HES as a robust, training-free metric that enables a unified, data-centric approach to efficiently developing advanced reasoning in LLMs.

## 1 INTRODUCTION

The ability of Large Language Models (LLMs) to solve complex problems through Chain-of-Thoughts (CoT) reasoning has become a central focus in frontier research (Jaech et al., 2024; DeepSeek-AI, 2025; Yang et al., 2025). To enhance models' reasoning capabilities, dominant training paradigms, including Supervised Fine-tuning (SFT) (Ouyang et al., 2022), Rejection Fine-tuning (RFT) (Yuan et al., 2023a), and Reinforcement Learning (RL) (Shao et al., 2024), heavily rely on high-quality training data. However, the indiscriminate expansion of training data often introduces more noise and incurs additional costs, spurring a critical need for efficient and effective data selection methods. The key lies in defining a robust metric to rapidly and accurately distinguish high-quality data from low-quality data (Li et al., 2024b).

This work focuses on the efficient training of reasoning models and pursues a lightweight, adaptive, and qualified selection algorithm for long-CoT responses. In long-CoT scenarios, the extended and thorough reasoning processes are more demanding and conducive to the model learning (Chen et al., 2025). While some approaches propose training additional task-specific models for selection (SHUM et al., 2025) or utilizing powerful LLMs to choose among multiple responses (Toshniwal et al., 2025), these methods come with significant computational costs, and the selected data is not dependent on the models being optimized, potentially leading to suboptimal training. Therefore, we aim to devise an efficient scoring mechanism that can automatically rank responses while optimizing both computational efficiency and methodological generality.

Currently, researchers have explored various strategies for filtering training data, such as length (Rae et al., 2021), perplexity (Marion et al., 2023), and average entropy (Sabbineni et al., 2023). However, these metrics demonstrate common limitations in long-CoT scenarios: they perform a coarse-grained, global evaluation of the entire reasoning path, treating all tokens with equal importance. This averaging approach overlooks the complex structure of the reasoning process, including planning, exploration and reflection (Chen et al., 2025). Within any given response, truly difficult-to-predict key tokens are typically in the minority, while the majority consists of easily predictable words. The averaging mechanism of traditional metrics dilutes the signal from these key tokens, making them challenging to identify genuinely high-quality data. This deficiency becomes even more pronounced in long-CoT scenarios involving extremely long sequences, where the few crucial tokens are likely overwhelmed by the excessive trivial tokens.

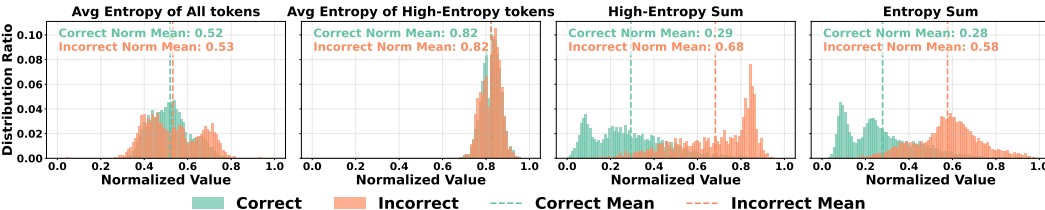

Figure 1: Comparative analysis of discriminative ability between HES and other metrics based on 512 responses per problem sampled by Qwen3-14B on AIME 2025. We can find that HES is most effective in distinguishing high- and low-quality samples.

To address this limitation, we shift from global evaluation to the identification of local, critical tokens. Wang et al. (2025) has suggested that a small number of high-entropy *forking tokens* in the reasoning process serve as a key driver for performance improvement. Inspired by this, we conduct a fine-grained analysis of token-level entropy throughout the reasoning process as shown in Figure 1. Here, an intuitive assumption is that a good reasoning process leads to correct results, while a poor one leads to incorrect results. Thus, we analyze the distributions of various entropy-based metrics for each group of correct and incorrect samples. Our findings confirm that average entropy struggles to distinguish between high- and low-quality samples. Even summing the entropy of all tokens is far less effective than summing the entropy of only the highest-entropy tokens. This reveals a strong correlation between the cumulative entropy of these key tokens and the quality of a sample.

Based on this observation, we introduce the High-Entropy Sum (HES), to quantify the complexity of a reasoning path. HES is calculated by summing the entropy values of the top 0.5% of tokens with the highest entropy. A higher HES score signifies a greater diversity and complexity of reasoning patterns, indicating a higher learning value. Leveraging this metric, we develop a unified and efficient data selection framework applicable across all major training paradigms.

Our experiments validate HES as a robust metric for effective and efficient data selection. In SFT, training on just the top 20% of samples ranked by HES achieves performance comparable to training on the full dataset; when the training data is expanded to the top 80%, model performance even consistently surpasses the full-dataset baseline. In RFT, its performance is significantly superior to traditional random selection, proving that HES can serve as an effective, training-free reward signal. Furthermore, in RL, we oversample a group of rollouts and then downsample top-50% of them ranked by HES as positive solutions. This strategy surpasses the normal setting where all rollouts are involved in the policy update.

In summary, our contributions are as follows:

- We introduce HES, an effective and efficient metric that measures reasoning quality by focusing on high-entropy tokens, overcoming the limitations of traditional metrics.
- We establish a unified data selection framework using HES that improves performance and efficiency across SFT, RFT, and RL, establishing it as a training-free reward signal.
- Our data-centric approach obviates the need for costly external reward models and provides a clear path toward building more powerful reasoning systems with greater efficiency.

## 2 PRELIMINARIES

### 2.1 TRAINING PARADIGMS

**Supervised Fine-Tuning (SFT).** Let $\mathcal{D} = \{(x, y)\}$ denote a corpus of correct demonstrations, where $x$ is a query and $y$ is the reference response. SFT minimizes the cross-entropy loss, defined by the objective function: $\mathcal{L}_{SFT}(\theta) = \mathbb{E}_{(x,y)\sim\mathcal{D}}[-\log \pi_\theta(y|x)]$, where $\theta$ represents the model's parameters and $\pi_\theta(y|x)$ is the probability assigned by the model to $y$ given $x$.

**Rejection Sampling Fine-Tuning (RFT).** RFT (Yuan et al., 2023a) augments SFT by generating training samples through the model's own exploration. The process involves three steps: (1) Generation: For query $x$, generate $m$ different candidate responses $\{y_1, y_2, \ldots, y_m\}$; (2) Selection: Use selection function $R(y)$ to select a response subset $Y$: $Y = \underset{y_i \in \{y_1, \ldots, y_m\}}{\arg\max} R(y_i)$; (3) Fine-tuning: Create new dataset $\mathcal{D}^*$ with $Y$ and fine-tune using SFT.

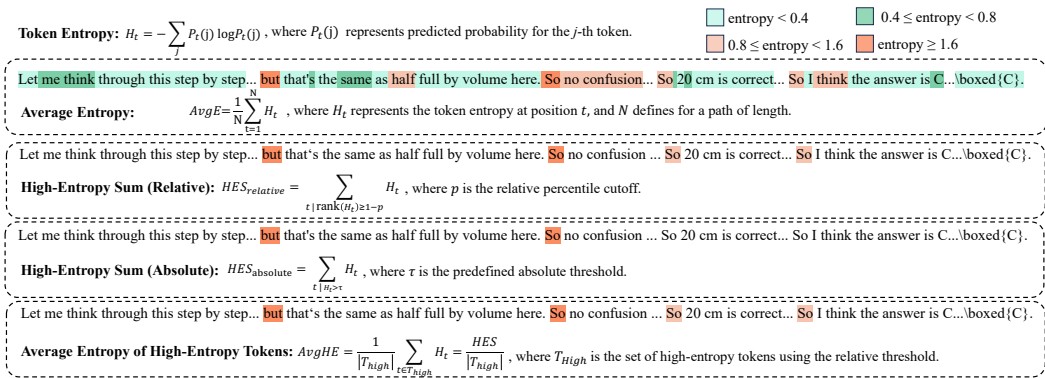

Figure 2: Different metrics calculation. The colored tokens are the tokens involved in the calculation.

**Reinforcement Learning (RL).** Group Relative Policy Optimization (GRPO) (Shao et al., 2024) is a variant of Proximal Policy Optimization (PPO) (Schulman et al., 2017). For each query, GRPO samples a group of $G$ responses $\{o_1, \ldots, o_G\}$ with corresponding rewards $\{r_1, \ldots, r_G\}$. The advantage $\hat{A}_i$ for response $o_i$ is first computed as: $\hat{A}_i = \frac{r_i - \text{mean}(\{r_j\}_{j=1}^G)}{\text{std}(\{r_j\}_{j=1}^G)}$. This group-normalized advantage is then used in a clipped policy gradient objective to update the model's parameters $\theta$. The full objective function to be maximized is: $\mathcal{J}_{\text{GRPO}}(\theta) = \mathbb{E}\left[\min\left(r_t(\theta)\hat{A}_i, \text{clip}(r_t(\theta), 1-\epsilon, 1+\epsilon)\hat{A}_i\right) - \beta D_{\text{KL}}(\pi_{\theta_{\text{old}}} \| \pi_\theta)\right]$. Here, $r_t(\theta)$ is the probability ratio between the new and old policies, $\hat{A}_i$ is the group-relative advantage, $\epsilon$ is a clipping hyperparameter, and the final term is a KL-divergence penalty with weight $\beta$ to stabilize training.

## 2.2 TOKEN ENTROPY

**Token entropy.** For a token position $t$ with probability distribution $P_t$ over the vocabulary, the token entropy $H_t$ is given by: $H_t = -\sum_j P_t(j) \log P_t(j)$, where $P_t(j)$ represents the predicted probability for the $j$-th token. Token entropy is the fundamental metric to measure a model's uncertainty during generation. Low entropy typically occurs during predictable parts of a reasoning path, such as completing a common phrase, performing a simple calculation, or following a standard template. Conversely, high entropy signifies high uncertainty where the model is considering multiple viable and often competing options. In the context of long-CoT reasoning, these high-entropy moments are particularly important as they often correspond to critical forks where the model must make a non-trivial decision that will shape the subsequent trajectory (Wang et al., 2025).

**Average Entropy.** The common approach to quantifying the overall uncertainty of a reasoning path is average entropy (Sabbineni et al., 2023), defined for a path of length $N$ as: $AvgE = \frac{1}{N}\sum_{t=1}^N H_t$. However, this global metric is limited because it masks critical local signals by averaging over long sequences. A high-quality reasoning path that successfully navigates multiple challenging forks may receive a similar score to one that follows a straightforward, low-complexity approach. This inability to distinguish between different levels of reasoning complexity makes the metric unreliable for identifying the most valuable training samples.

## 3 QUANTIFYING REASONING QUALITY VIA HIGH-ENTROPY SUM

To overcome the limitations of global metrics, we first introduce HES as a quantitative measure of reasoning quality and then leverages it to guide data selection across major training paradigms.

## 3.1 HIGH-ENTROPY SUM

Our core metric is HES, which quantifies the cumulative intensity of pivotal, high-entropy moments in a reasoning path. We define this metric in two primary formulations: a robust, adaptive version using a relative, percentile-based threshold ($HES_{relative}$), and a simpler alternative using a fixed, absolute threshold($HES_{absolute}$).

$HES_{relative}$.    This metric is designed to capture the cumulative complexity of a sample by focusing on its most uncertain tokens in an adaptive manner.

$$HES_{relative} = \sum_{t \,|\, \mathrm{rank}(H_t) \geq 1-p} H_t. \tag{1}$$

Here, $\mathrm{rank}(H_t)$ selects the tokens $t$ whose entropy $H_t$ rank within the top $p$-th percentile of a sample for summation (e.g., $p = 0.005$ for the top 0.5%). A higher $HES_{relative}$ score indicates the successful navigation of more numerous and intense forks, which represents higher quality. The relative threshold makes this metric robust to variations in length among different models and tasks.[1]

$HES_{absolute}$.    Alternatively, this metric is used for quality estimation by a fixed cutoff.

$$HES_{absolute} = \sum_{t \,|\, H_t > \tau} H_t. \tag{2}$$

Here, the sum is taken over the tokens $t$ whose entropy $H_t$ exceeds a predefined absolute value $\tau$ (e.g., $\tau = 1.6$). While less adaptive than the relative threshold, this method provides a straightforward alternative for quality estimation, particularly in contexts where a consistent entropy scale can be expected across all samples.

$AvgHE$.    To test whether the cumulative sum of uncertainty is a more effective signal than its average intensity, we introduce average entropy of high-entropy tokens.

$$AvgHE = \frac{1}{|T_{high}|} \sum_{t \in T_{high}} H_t = \frac{HES}{|T_{high}|}. \tag{3}$$

For this metric, we identify the set of high-entropy tokens $T_{high}$ using the relative threshold described above. $AvgHE$ is defined as $HES_{relative}$ normalized by $|T_{high}|$, the number of tokens in the set. It is designed to isolate the average complexity of key-fork tokens, different from $AvgHE$.

## 3.2 HES-Guided Data Selection

We apply the metrics defined above to curate data and guide training across three major paradigms.

**SFT.**    We evaluate HES by training on several curated subsets created by ranking the entire dataset. We train on the top 20% of highest-HES samples to test for sample efficiency, and on the top 80% of highest-HES samples to test whether pruning the lowest-quality data can surpass full-dataset performance. As a control, we also train on the bottom 20% of lowest-HES samples to measure the impact of low-quality data. To validate our specific formulation, we compare against models trained on the top 20% of samples selected by other metrics, including $HES_{absolute}$, $AvgHE$ and $AvgE$.

**RFT.**    First, we generate multiple candidate responses for each query and filter this pool to retain only those that yield the correct answer. Our selection process is then divided into per-query and global-pool selection. In the former, we select the candidates with the highest HES from the response set for each query. This setting tests the ability of HES to identify the high-quality solutions from a small, localized pool of candidates for a specific problem. In the latter, we aggregate all query-response pairs and select the candidates with the highest HES from this entire pool. Finally, we use this curated dataset to fine-tune the model. This more challenging setting tests the ability of HES to distinguish high-quality responses from a diverse, global distribution.

**RL.**    We design a novel asymmetric sampling method based on HES. Our strategy involves two key components: for positive reinforcement, we select half with the highest HES from the pool of successful trajectories, making the model to deconstruct and master the most complex correct solutions. And for negative feedback, we randomly sample half failures, ensuring the model is exposed to a diverse and unbiased distribution of errors to enhance its overall robustness.

---

[1]As experiments will demonstrate, $HES_{relative}$ provides better results than $HES_{absolute}$. This is because its adaptive nature makes it more robust across the diverse entropy distributions found in different models and reasoning paths. Consequently, for the remainder, $HES$ will refer to $HES_{relative}$ unless otherwise specified.

# 4 EXPERIMENTS

## 4.1 SFT EXPERIMENTS

### 4.1.1 EXPERIMENTAL SETUP

**Datasets and Models.** Our training data is sourced from Open-Math-Reasoning (Moshkov et al., 2025) and Open-R1-220k (Hugging Face, 2025). The former contains 3.2 million samples, and we sample approximately 100,000 examples from the CoT portion for our experiments. For the latter, we utilize the "default" subset, which includes 94,000 math problems and their corresponding solutions. Detailed description of these datasets is given in the Appendix. We conduct experiments on Qwen3-8B-Base (Yang et al., 2025) and DeepSeek-R1-Distilled-7B (DeepSeek-AI, 2025). These models are chosen for their strong long-CoT capabilities and because they are fully open-source, ensuring the reproducibility.

**Evaluation Settings.** We evaluate all the checkpoints with the best performance of SFT, RFT and RL experiments on a suite of seven challenging, open-source benchmarks: AIME24 (Art of Problem Solving, 2024a;b), AIME25 (Art of Problem Solving, 2025a;b), HMMT23 (HMMT, 2023),HMMT24 (HMMT, 2024), HMMT25 (HMMT, 2025), OlymMATH (Sun et al., 2025), and GPQA (Rein et al., 2024). The first six are mathematical competition, while GPQA consists of graduate-level STEM tasks. For all benchmarks, we report the average pass@1 accuracy over 16 sampling paths per problem, using a temperature of 0.6 and a maximum generation length of 32,768.

**Training Details.** Our implementation is based on the `open-r1` (Hugging Face, 2025) framework, and we follow the recommended SFT hyperparameters. We use the AdamW optimizer with a learning rate of $4 \times 10^{-5}$. The learning rate follows a cosine decay schedule with a warm-up ratio of 0.1. All experiments are run with a global batch size of 64 and are trained for a total of 3 epochs.

**Experimental Design.** To comprehensively evaluate the effectiveness of HES, we conduct experiments with the following settings: (1) **Full-Dataset**: Training on the complete dataset as performance upper bound. (2) **Random**: Training on random subsets as baseline. (3) **Highest-Difficulty**: For Open-Math-Reasoning, difficulty scores are derived from pass@32 using Qwen2.5-Math-72B-Instruct. For Open-R1-220k, scores are based on the number of correct answers obtained from 4 sampling attempts using DeepSeek-R1. And we select the most difficult samples. (4) **Medium-Difficulty**: We utilize the same difficulty scoring mechanism as defined above. However, instead of selecting the most difficult samples, we select those with intermediate scores (specifically, samples ranking within the middle percentile range of the difficulty distribution), filtering out both trivial and extremely hard cases. (5) **Length**: Selecting the longest samples. (6) **Forking-Only**: Applying gradient updates only to the high-entropy tokens (Wang et al., 2025). (7) **Highest-AvgE**: Selecting by the highest average entropy across all tokens. (8) **Highest-AvgHE**: Selecting by the highest average entropy of high-entropy tokens. (9) **Highest-ES**: Selecting by the highest total entropy sum. (10) **Highest-HES**$_{\text{absolute}}$: Training on the samples with highest HES using fixed absolute threshold. (11) **Lowest-HES**: Training on samples with the lowest HES. (12) **Highest-HES**: Similar to (9) but using relative threshold.

### 4.1.2 MAIN RESULTS

**HES outperforms alternative selection methods.** Our primary finding is that HES serves as a superior signal for identifying high-quality training data compared to all baselines. As shown in Table 1, training on the Highest-HES-20% subset not only significantly outperforms the Random-20% baseline but also closely approaches the performance of training on the Full-Dataset with only a fifth of the data. Notably, in Table 2, this 20% subset selected by Highest-HES even surpasses the performance of training on the entire dataset. Furthermore, HES proves to be a more effective indicator of quality than other entropy-based metrics. It surpasses models trained on data selected by the average entropy of high-entropy tokens, the global average entropy across all tokens, and the total sum of all token entropy. It also shows a clear advantage over simpler heuristics like length and difficulty. Additionally, a relative threshold is superior for selecting high-entropy tokens because it normalizes for diverse entropy distributions across different models and reasoning paths, making it more robust than a fixed, absolute threshold. This establishes HES with relative threshold as the most robust metric for identifying the most high-quality subset of the data for training.

| Method | Ratio | AIME24 | AIME25 | HMMT23 | HMMT24 | HMMT25 | Oly(E) | Oly(H) | GPQA | AVG |
|---|---|---|---|---|---|---|---|---|---|---|
| Full-Dataset | 100 | 50.83 | 34.17 | 35.21 | 28.13 | 24.58 | 42.94 | **6.94** | 38.04 | 32.61 |
| Random | 20 | 39.79 | 27.92 | 24.79 | 20.21 | 20.42 | 30.00 | 3.88 | 40.09 | 25.89 |
| Random | 80 | 47.92 | 36.46 | 34.58 | 25.21 | 25.00 | 43.00 | 6.69 | 37.63 | 32.06 |
| Highest-Difficulty | 20 | 45.83 | 34.17 | 30.00 | 25.63 | 22.50 | 35.38 | 5.88 | 39.68 | 29.88 |
| Medium Difficulty | 20 | 38.13 | 27.50 | 25.83 | 22.50 | 18.96 | 26.50 | 3.69 | 40.40 | 23.29 |
| Length | 20 | 46.67 | 34.79 | 31.04 | 24.79 | 23.54 | 36.06 | 6.44 | 42.05 | 30.67 |
| Forking-Only | 100 | 52.92 | 36.04 | 35.00 | 28.33 | 22.71 | 42.31 | 6.75 | 35.98 | 32.51 |
| Highest-AvgE | 20 | 40.21 | 31.25 | 28.54 | 23.13 | 19.17 | 31.19 | 4.88 | 40.85 | 27.40 |
| Highest-AvgHE | 20 | 42.29 | 29.79 | 31.46 | 22.08 | 18.75 | 33.06 | 4.31 | 42.01 | 27.97 |
| Highest-ES | 20 | 47.92 | 34.17 | 30.42 | 25.63 | 22.71 | 38.50 | 5.25 | 42.78 | 30.92 |
| Lowest-HES | 20 | 18.54 | 18.96 | 11.88 | 11.04 | 7.92 | 11.00 | 2.94 | 36.90 | 14.90 |
| Highest-HES$_{absolute}$ | 20 | 42.29 | 32.29 | 32.92 | **29.38** | 19.38 | 36.06 | **6.94** | 41.64 | 30.11 |
| Highest-HES | 20 | 47.29 | 36.04 | 33.13 | 25.63 | 22.08 | 36.31 | 5.56 | 43.06 | 31.14 |
| Highest-HES (0.6B) | 20 | 49.17 | 35.42 | 32.50 | 24.17 | 22.29 | 41.50 | 6.38 | **45.55** | 32.12 |
| Highest-HES (1.7B) | 20 | 47.08 | 34.79 | 31.04 | 26.04 | 21.46 | 40.06 | 4.56 | 45.20 | 31.28 |
| Highest-HES | 80 | **56.67** | **41.88** | **37.92** | 27.08 | **26.88** | 46.31 | 6.94 | 39.24 | **35.36** |

Table 1: Performance comparison of SFT using Qwen3-8B-Base on Open-Math-Reasoning. "Ratio" indicates the percentage of the full dataset for training. Results are shown as average@16(%). Oly(E) and Oly(H) denote the easy and hard subsets of OlymMATH respectively. **Bold** indicates the best performance per benchmark. "AVG" represents the mean performance across all benchmarks.

| Method | Ratio | AIME24 | AIME25 | HMMT23 | HMMT24 | HMMT25 | Oly(E) | Oly(H) | GPQA | AVG |
|---|---|---|---|---|---|---|---|---|---|---|
| Full-Dataset | 100 | 46.25 | 33.13 | 31.67 | 26.25 | 22.29 | 38.50 | 5.50 | 38.13 | 30.22 |
| Random | 20 | 42.08 | 37.08 | 29.79 | 26.67 | 23.13 | 38.13 | 5.13 | 41.04 | 30.38 |
| Highest-HES | 20 | **51.67** | **40.83** | 33.96 | **30.21** | **24.58** | **45.75** | **6.88** | **42.99** | **34.61** |
| Highest-HES | 80 | 50.42 | 35.21 | **35.21** | 26.25 | 24.17 | 42.25 | 5.44 | 39.87 | 32.35 |
| Lowest-HES | 20 | 29.38 | 27.29 | 17.50 | 18.33 | 14.58 | 19.25 | 3.06 | 36.84 | 20.78 |

Table 2: Performance comparison of SFT using DeepSeek-R1-Distilled-Qwen-7B on OpenR1-Math-220k. **Bold** indicates the best performance per benchmark.

| Method | Ratio | LiveBench | AIME25 | GPQA | AVG |
|---|---|---|---|---|---|
| Fullset | 100 | 58.76 | 20.00 | 30.08 | 36.28 |
| Random | 20 | 49.06 | 22.29 | 35.45 | 35.60 |
| Highest-HES | 20 | 54.59 | **25.00** | **39.02** | **39.54** |
| Highest-HES | 80 | **61.38** | 23.54 | 33.62 | 39.51 |
| Lowest-HES | 20 | 35.32 | 12.92 | 29.36 | 25.86 |

Table 3: SFT performance on **Code Domain**.

| Method | Ratio | MMLU | GPQA | HMMT25 | AVG |
|---|---|---|---|---|---|
| Fullset | 100 | 85.38 | 47.66 | 0.21 | 44.42 |
| Random | 20 | 85.82 | 45.86 | 5.63 | 45.77 |
| Highest-HES | 20 | **88.77** | **50.95** | **8.96** | **49.56** |
| Highest-HES | 80 | 86.08 | 49.75 | 0.63 | 45.48 |
| Lowest-HES | 20 | 74.88 | 31.44 | 0.21 | 35.51 |

Table 4: SFT performance on **STEM Domain**.

**Removing low-quality data boosts performance.** The model, trained the 80% highest-HES of samples by simply pruning the 20% with the lowest HES, achieves an average accuracy of 35.36%. This result not only surpasses Random-80% but also consistently and significantly outperforms Full-Dataset. This counter-intuitive result strongly suggests that the lowest-quality samples are not merely uninformative but act as training noise, and their removal provides a performance benefit. This conclusion is further supported by the performance of the Lowest-HES-20%. It average score is only 14.90% — far below even Random-20%. This confirms that these lowest-HES samples are indeed harmful to model training and HES is highly effective at identifying them. It further proves that data quality is more important than data quantity (Li et al., 2024b).

**HES is a consistent metric across models and datasets.** To ensure the robustness of HES, we replicate our key experiments across different models and datasets, where the core trends remain highly consistent. The principle that pruning low-quality data improves performance is validated across both setups. Specifically, Highest-HES-80%(Table 2) surpasses Full-Dataset on OpenR1-Math-220k. This consistency validates HES as a reliable metric for identifying high-quality data.

**HES generalizes effectively to non-mathematical domains.** To verify the universality of HES beyond mathematics, we extended our evaluation to Code Generation and Scientific Reasoning domains. We utilized the codeforces-cots (Hugging Face, 2025) dataset for code tasks and the STEM

subset of the Llama-Nemotron-Post-Training-Dataset (Bercovich et al., 2025) for scientific tasks. The core trends remain highly consistent across these setups. The principle that pruning low-quality data improves performance is validated across both new domains. Specifically, Highest-HES-80% consistently surpasses the Full-Dataset baseline in the Code domain (Table 3), and this performance gain is also replicated in the STEM domain (Table 4). Moreover, training on just the top 20% of data ranked by HES yields even more significant improvements, surpassing the full dataset by over 3% and 5% respectively. These results confirm that HES captures intrinsic reasoning quality signals common across diverse logic-intensive tasks, supporting its role as a unified data selection metric.

**HES enables efficient small-to-large model transfer.** To assess the transferability of our metric, we utilized smaller, computationally efficient models (e.g., Qwen3-0.6B) to screen data for training the larger Qwen3-8B model. Remarkably, the performance achieved using the 0.6B proxy model (Avg 32.12%) is comparable to that of the 8B model's self-selection (Avg 31.14%), while reducing inference costs by over an order of magnitude. This strong cross-model consistency suggests that HES captures intrinsic reasoning complexity inherent to the data, rather than model-specific artifacts, making it a highly cost-effective strategy for large-scale data curation.

## 4.2 RFT Experiments

### 4.2.1 Experimental Setup

**Dataset and Model.** For RFT, we use the DeepScaleR dataset (Luo et al., 2025a), which comprises 40,000 question-answer pairs. The experiments are conducted on the DeepSeek-R1-Distilled-Qwen-7B.

**Training Details.** First, we sample 32 candidate responses using the model for each query in the dataset. Then, we filter these candidates to retain only the correct trajectories. This creates a pool which serves as the foundation for selection. Training settings, such as the optimizer and learning rate schedule, remain consistent with SFT, with the only difference being the curated training dataset.

**Experimental Design.** Our experiments are divided into two settings at both a local and global level. **(1) Per-Query Selection.** We construct training datasets by selecting a fixed number of correct responses ($k \in \{2, 4, 8\}$) for each query. If a given question has fewer than $k$ available correct responses, all of them are used. **(2) Global Pool Selection.** We first aggregate all correct responses from all queries into a single, large pool. We then construct training datasets by sampling responses from this global pool, where the total sampled number equals to the dataset size in the corresponding per-query setting.

Within each of these two settings, we compare the following strategies: **(1) Random:** Randomly sample the required number of responses. **(2) Lowest-HES:** Select the responses with the lowest HES. **(3) Highest-HES:** Select the responses with the highest HES. **(4) Length:** Select the responses with the longest token count. **(5) Difficulty:** Select the responses based on difficulty scores (e.g., medium difficulty).

### 4.2.2 Main Results

**HES shows robust performance in both Per-Query and Global Pool settings.** The superiority of HES remains robust across both the localized per-query and the more challenging global pool setting, consistently outperforming strong heuristic baselines such as length and difficulty. Notably, in the global pool setting, strategies relying solely on length or difficulty failed to surpass the random baseline, whereas HES achieved significant gains. An interesting finding is that the per-query selection often yields better absolute performance than the corresponding global pool selection. We hypothesize this is because the per-query approach guarantees that every query is represented in the final training data, thus preserving query diversity. This suggests that while HES is a powerful quality metric, a selection strategy that maintains query diversity is also crucial for maximizing model performance.

**HES shows robust performance in both Per-Query and Global Pool settings.** The superiority of HES remains robust across both the localized per-query and the more challenging global pool

| Method | AIME24 | AIME25 | HMMT23 | HMMT24 | HMMT25 | Oly(E) | Oly(H) | GPQA | AVG |
|---|---|---|---|---|---|---|---|---|---|
| *Per-Query Selection ($k = 2$)* | | | | | | | | | |
| Random | 46.46 | 34.17 | 31.88 | **28.54** | 21.46 | 35.63 | 4.31 | 40.50 | 30.37 |
| Highest-HES | **48.33** | **34.58** | **34.38** | 28.13 | **21.67** | **38.06** | **5.56** | 40.30 | **31.38** |
| Lowest-HES | 43.75 | 34.17 | 30.63 | 25.00 | 18.96 | 33.81 | 4.44 | 39.43 | 28.77 |
| Length | 46.04 | 33.33 | 33.75 | 28.13 | 19.58 | 35.94 | 4.81 | **40.56** | 30.27 |
| *Global Pool Selection ($k = 2$)* | | | | | | | | | |
| Random | 42.29 | 31.04 | 30.00 | **26.88** | 18.54 | 30.81 | 4.75 | 38.35 | 27.83 |
| Highest-HES | **46.46** | 34.58 | **32.50** | 25.42 | **20.83** | **34.00** | 5.31 | 42.30 | **30.18** |
| Lowest-HES | 19.38 | 14.58 | 10.21 | 11.04 | 7.92 | 10.81 | 2.00 | 31.19 | 13.39 |
| Difficulty | 45.00 | 32.71 | 28.13 | 23.54 | 17.71 | 30.13 | 4.31 | 36.93 | 27.31 |
| Length | 45.21 | **35.42** | 27.92 | 25.21 | 18.33 | 32.13 | 5.13 | 39.55 | 28.61 |
| *Per-Query Selection ($k = 4$)* | | | | | | | | | |
| Random | 48.33 | 31.88 | 33.33 | 27.71 | 18.75 | 36.00 | 4.50 | 38.26 | 29.85 |
| Highest-HES | **48.96** | **36.25** | **33.75** | **29.17** | 21.04 | **37.13** | **5.13** | 40.85 | **31.54** |
| Lowest-HES | 43.75 | 33.54 | 30.42 | 26.46 | **21.67** | 32.69 | 4.75 | 36.74 | 28.75 |
| Length | 46.04 | 33.33 | 33.75 | 28.13 | 19.58 | 35.94 | 4.81 | 40.44 | 30.25 |
| *Global Pool Selection ($k = 4$)* | | | | | | | | | |
| Random | 45.42 | **38.13** | 30.83 | **26.04** | 20.21 | 33.06 | 4.63 | 37.12 | 29.43 |
| Highest-HES | **48.96** | 33.13 | 30.00 | 24.38 | **21.88** | **36.19** | **5.56** | 42.52 | **30.33** |
| Lowest-HES | 21.46 | 18.96 | 12.92 | 11.04 | 9.38 | 11.56 | 1.69 | 30.52 | 14.69 |
| Difficulty | 45.00 | 32.71 | 28.12 | 23.54 | 17.71 | 30.13 | 4.31 | 33.27 | 26.85 |
| Length | 45.21 | 35.42 | 27.92 | 25.21 | 18.33 | 32.13 | 5.13 | 38.32 | 28.46 |
| *Per-Query Selection ($k = 8$)* | | | | | | | | | |
| Random | 49.17 | 34.58 | 29.38 | 28.75 | **22.08** | 36.69 | 4.69 | 35.95 | 30.16 |
| Highest-HES | **50.42** | **37.92** | 30.42 | 28.96 | **22.08** | 35.31 | **5.81** | 38.10 | **31.13** |
| Lowest-HES | 42.50 | 32.50 | **32.71** | 28.54 | 20.00 | 34.69 | 4.88 | 34.63 | 28.81 |
| Length | 46.67 | 35.42 | 32.29 | **30.21** | 20.63 | 35.94 | 5.63 | **38.57** | 30.67 |
| *Global Pool Selection ($k = 8$)* | | | | | | | | | |
| Random | 46.67 | 35.42 | 31.67 | 28.33 | 18.54 | 33.06 | 4.19 | 35.61 | 29.19 |
| Highest-HES | **48.96** | **36.46** | **31.88** | **29.17** | 19.38 | **38.31** | 5.19 | 39.20 | **31.07** |
| Lowest-HES | 25.63 | 21.25 | 18.96 | 14.66 | 11.67 | 14.50 | 2.44 | 30.46 | 17.45 |
| Difficulty | 44.38 | 32.29 | 31.04 | 24.79 | 20.42 | 31.50 | 4.63 | 33.36 | 27.80 |
| Length | 48.75 | 36.04 | 30.42 | 26.46 | **22.08** | 36.44 | **5.56** | 37.82 | 30.45 |

Table 5: Performance comparison of RFT experiments across different candidate pool sizes ($k \in \{2, 4, 8\}$). **Bold** indicates the best result for each benchmark within each setting.

setting. An interesting finding is that the per-query selection often yields better absolute performance than the corresponding global pool selection. We hypothesize this is because the per-query approach guarantees that every query is represented in the final training data, thus preserving query diversity. This suggests that while HES is a powerful quality metric, a selection strategy that maintains query diversity is also crucial for maximizing model performance.

**The effectiveness of HES is maintained as the candidate pool size increases.** We show that the utility of HES as a selection signal is robust as the number of candidate responses per prompt ($k$) increases from 2 to 8. While the performance generally improves with a larger candidate pool, the advantage of the Highest-HES strategy over the Random is consistently maintained. The average performance gain in the per-query setting is +1.01 points for $k = 2$, +1.69 points for $k = 4$, and +0.97 points for $k = 8$. This non-monotonic but consistently positive trend suggests that while there may be diminishing marginal returns on the number of responses, the value of using a principled selection metric like HES remains significant. This confirms the utility of HES in sample-rich environments where an efficient and accurate selection signal is needed.

## 4.3 RL EXPERIMENTS

### 4.3.1 EXPERIMENTAL SETUP

**Dataset and Model.** Following DeepScaleR-1.5B-Preview (Luo et al., 2025a), we conduct our RL experiments by training DeepSeek-R1-Distilled-Qwen-1.5B (DeepSeek-AI, 2025) on DeepScaleR.

| Method | AIME24 | AIME25 | HMMT23 | HMMT24 | HMMT25 | Oly(E) | Oly(H) | GPQA | AVG |
|---|---|---|---|---|---|---|---|---|---|
| Full-Batch | 33.33 | 25.63 | 17.30 | 14.00 | **15.21** | 19.69 | 3.19 | **36.71** | 20.63 |
| Pos-Rand, Neg-Rand | 32.30 | 25.21 | 13.54 | 14.11 | 13.96 | 19.18 | 4.25 | 36.49 | 19.88 |
| Pos-Rand, Neg-Low | 33.75 | 24.79 | 15.42 | 13.73 | 11.67 | 19.00 | 3.38 | 36.30 | 19.76 |
| Pos-Low, Neg-Rand | 30.42 | 24.17 | 15.63 | 13.10 | 14.17 | 19.50 | 3.25 | 34.97 | 19.40 |
| Pos-High, Neg-Low | 31.25 | 25.42 | 15.21 | 13.96 | 12.71 | 18.88 | 3.56 | 35.01 | 19.50 |
| Pos-Difficulty, Neg-Rand | 35.00 | 24.79 | 16.88 | 13.77 | 14.17 | 18.94 | 3.75 | 34.88 | 20.27 |
| Pos-Longest, Neg-Rand | 31.04 | 26.25 | 16.46 | 13.95 | 14.79 | **20.00** | 4.06 | 35.32 | 20.23 |
| **Pos-High, Neg-Rand** | **35.42** | **27.29** | **17.92** | **18.13** | 11.88 | 19.88 | **4.31** | 35.54 | **21.30** |

Table 6: RL results comparing different sampling strategies. All reported values are average@16(%). The best result in each column is in **bold**.

**Training Details.** Our implementation is built upon the `verl` codebase (Sheng et al., 2025) and follows the standard GRPO settings. Consistent with the DeepScaleR-1.5B-Preview (Luo et al., 2025a) setup, the maximum generation length is set to 8192 tokens. We use a temperature of 0.6 and generate 32 rollouts per query. We train for 3 epochs, a total of 628 steps. It is worth noting that we train the baseline to its officially reported accuracy.

**Experimental Design.** For each query, we first generate a pool of 32 rollouts and separate them into positive (correct) and negative (incorrect) pools. We compare the performance of using all rollouts against several down-sampling strategies that select half of the trajectories. The strategies include: **(1) Full-Batch:** We use all 32 generated rollouts for GRPO update without selection. **(2) Pos-Rand, Neg-Rand:** We randomly select half of both positive and negative trajectories to maintain original distribution. **(3) Pos-High, Neg-Rand:** We select positive trajectories with the highest HES, paired with random negative samples. **(4) Pos-Rand, Neg-Low:** We pair random positive samples with negative trajectories having the lowest HES. **(5) Pos-High, Neg-Low:** We combine the highest-HES positive trajectories with the lowest-HES negative trajectories for maximal contrast. **(6) Pos-Low, Neg-Rand:** An ablation using the lowest-HES positive trajectories with random negative samples. **(7) Pos-Difficulty, Neg-Rand:** We select positive trajectories based on difficulty scores, paired with random negative samples. **(8) Pos-Longest, Neg-Rand:** We select positive trajectories with the longest token count, paired with random negative samples.

### 4.3.2 MAIN RESULTS

**Focusing on high-quality positive samples unlocks superior performance and efficiency.** Our primary finding is that a targeted, asymmetric sampling strategy—selecting for the highest-HES positive rollouts while pairing them with randomly sampled negative rollouts—unlocks superior performance. As detailed in Table 6, this Pos-High, Neg-Rand strategy achieves the highest average accuracy of 21.30%. In contrast, alternative strategies selecting positive samples based on difficulty or length, while slightly better than random sampling, failed to match the Full-Batch baseline. Most notably, HES is the unique strategy that not only significantly outperforms the standard down-sampling baseline (Pos-Rand, Neg-Rand, 19.88%) but also surpasses the performance of the Full-Batch baseline, despite using only half the training data per update step. This result provides strong evidence that a smaller, more targeted training batch focused on high-quality positive samples is more effective and capital-efficient than a larger, undifferentiated one.

**The results underscore the critical role of negative sample diversity.** An equally important insight comes from the performance degradation of strategies that curate the negative pool. The results in Table 6 shows that both Pos-Rand, Neg-Low (19.76%) and Pos-High, Neg-Low (19.50%) strategies perform worse than the simple Pos-Rand, Neg-Rand (19.88%). This suggests that while HES is a powerful tool for selecting high-quality positive samples, constraining the negative ones to be overly simplistic is useless. It is more effective to expose the model to a diverse distribution of failure modes, as provided by random sampling, which can build robust and generalizable reasoning.

### 4.4 SENSITIVITY ANALYSIS

To validate the robustness of HES, as shown in Figure 3 and 4, we analyze its sensitivity to two key hyperparameters: the data selection ratio and the high-entropy token ratio used in our SFT experiments across Math, STEM and Code domain. First, across all benchmarks, peak performance is almost consistently achieved when using approximately 20% or 80% of the data as ranked by HES,

confirming that pruning the samples with the lowest quality is a reliable strategy for improvement. Second, while HES is robust to the specific high-entropy token ratio used, our experiments show that smaller, more targeted ratios of 0.005 and below consistently deliver the best performance.

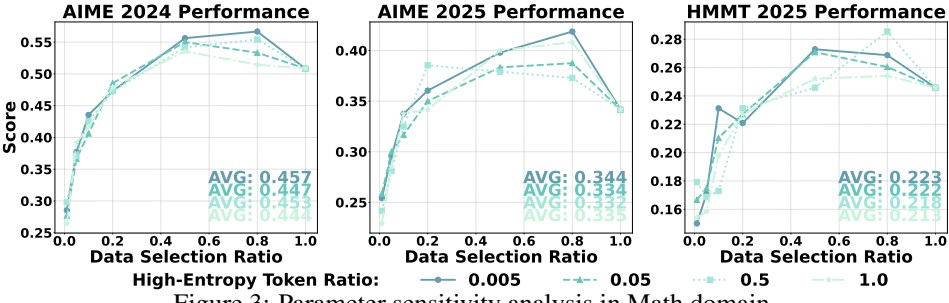

Figure 3: Parameter sensitivity analysis in Math domain.

Figure 4: Parameter sensitivity analysis in STEM and Code domain.

## 5  RELATED WORK

**Training-Time Efficiency.**  Improving training-time efficiency is a critical challenge across the dominant paradigms for LLM reasoning: SFT, RFT, and RL. To address this, some works focus on curating higher-quality training sets through methods like verification-based selection or preference optimization (Hu et al., 2022; Rein et al., 2024; Yang et al., 2025; Yuan et al., 2023b). Other works aim to provide more granular learning signals by directly supervising intermediate steps with process-level rewards (Lightman et al., 2023; Cui et al., 2025; Luo et al., 2025b). While powerful, the effectiveness of these methods hinges on a reliable signal to identify which reasoning paths or steps are valuable, and they often rely on costly external reward models or detailed human annotations. Our work addresses this need by introducing HES, a simple, intrinsic metric that provides this critical signal uniformly across all three paradigms.[2]

**Data Selection.**  Data selection aims to identify optimal training subsets from large-scale datasets (Albalak et al., 2024; Li et al., 2024a).  Traditional approaches rely on heuristic rules (Robertson et al., 2009; Xie et al., 2023) or global metrics like perplexity (Wettig et al., 2024; Marion et al., 2023). While gradient-based methods offer finer-grained selection (Killamsetty et al., 2021; Han, 2023), their computational costs scale prohibitively with sequence length and model size. Alternative approaches using task-specific models (SHUM et al., 2025) or LLM-based selection (Toshniwal et al., 2025) introduce substantial overhead. Current data selection thus faces two challenges: (1) the inability to identify crucial tokens through global metrics, and (2) the lack of efficient metrics deployable across different training paradigms. Our HES addresses both limitations.

## 6  CONCLUSION

In this work, we introduce HES, an effective and efficient metric that measures reasoning quality by focusing on key high-entropy tokens.  Validated across SFT, RFT, and RL, our HES-guided framework enables efficiently training more robust models with greater performance. By leveraging the simple metric to obviate the need for costly external reward models, this work establishes a data-centric path toward the development of next-generation reasoning systems.

---

[2]A complete discussion of related work is given in the Appendix.

## REPRODUCIBILITY STATEMENT

To ensure the reproducibility of our research, we have provided a detailed description of all experimental settings, including models, datasets, and training hyperparameters, in the Experiments section of this paper. Our work is built exclusively on publicly available, open-source models and datasets to facilitate replication. We are committed to releasing all of our data curation scripts and training code after the paper is accepted.

## ETHICS STATEMENT

All experiments were conducted using publicly available, open-source LLMs. The datasets used for training and evaluation are also publicly available or derived from well-established public benchmarks. We have taken care to decontaminate our training data to prevent overlap with our evaluation benchmarks, ensuring a fair and rigorous assessment of our methods. The primary goal of this work is to develop more efficient and principled methods for training LLMs. By enabling the development of more powerful models with less data and computational cost, we aim to make advanced AI research more accessible and sustainable. We acknowledge the potential dual-use nature of powerful language models and have focused our research on foundational reasoning capabilities, not on developing applications that could cause societal harm. We are committed to transparency within the research community.

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

## A    STATEMENT ON THE USE OF LLMS

LLMs referenced and evaluated in this work are the primary subject of our research. All experiments, analyses, and results were conceived and conducted by the human authors. No language models were used for the generation of core research ideas, experimental design, or the substantive text of this manuscript. The authors take full responsibility for the entire content of this paper, including the accuracy of all claims and the final wording of the text.

## B    RELATED WORK

**Efficient Reasoning.**    Recent progress on efficient reasoning has advanced along two directions: (1) test-time efficiency (Jaech et al., 2024; Snell et al., 2024) and (2) training-time efficiency. In the former, mainstream methods broaden the explored trajectory set by deepening CoT and branching into multiple paths (Wei et al., 2022; Sessa et al., 2024; Yao et al., 2023; Besta et al., 2024; Sel et al., 2023; Ning et al., 2023; Fu et al., 2025). Building on this, some work incorporates tool use, heuristic control, or self-reflection to verify processes or outcomes and stabilize search (Chen et al., 2022; Gou et al., 2023; Yu et al., 2025; Renze & Guven, 2024). However, these methods often require substantial token budgets. This motivates efforts to compress outputs by extracting key information from reasoning trajectories (Zhang et al., 2025; Xu et al., 2025; Aytes et al., 2025). On the training-time side, the focus is on efficiently converting exploration-derived signals into policy updates. Dominant pipelines improve reasoning by verification-based selection or preference optimization (Hu et al., 2022; Rein et al., 2024; Yang et al., 2025; Yuan et al., 2023b). In addition, some work explicitly targets process quality by supervising intermediate steps with process-level signals (Lightman et al., 2023; Cui et al., 2025; Luo et al., 2025b). Therefore, efficient reasoning either allocates more computation to critical branches at test time or concentrates training budgets on samples or spans with higher pedagogical value at training time. Both require a stable signal that reliably identifies critical fork tokens. Our work is mainly focused on training-time efficiency, and we leave the exploration of test-time efficiency to future work.

**Data Selection.**    Data selection takes the full training data as input and chooses a subset to train (Albalak et al., 2024). Large-scale, high-quality datasets play a decisive role in training effectiveness (Li et al., 2024a; Penedo et al., 2024). Traditional selection methods rely on rules and heuristics, which make it difficult to assess the quality and complexity of reasoning processes (Robertson et al., 2009; Xie et al., 2023). Therefore, many works adopt global proxies such as perplexity, average token entropy, sample length, or heuristic difficulty (Wettig et al., 2024; Marion et al., 2023; Ivison et al., 2022). Yet these metrics inevitably mix routine low-entropy spans with the sparse high-entropy tokens that signal critical decision points, which dilutes the signal and makes it hard to distinguish correct solutions with low process complexity from those with high process complexity. More fine-grained strategies rely on selection criteria derived from gradients or influence estimates (Killamsetty et al., 2021; Han, 2023; Xia et al., 2024). However, their compute and memory costs scale rapidly with sequence length and model size, which limits their practicality for large-scale training. Consequently, current data selection faces two core gaps: (1) the lack of a model-intrinsic signal that directly captures process-level critical decision points; (2) the lack of a metric that can be deployed consistently and efficiently across diverse training methods without introducing new supervision. Our proposed HES addresses both gaps.

## C    LIMITATIONS

While our findings demonstrate the effectiveness of HES, we acknowledge several limitations that provide avenues for future research. Our experiments are primarily conducted on mathematical reasoning, and the generalizability of HES as a quality proxy to other complex domains, such as code generation, remains an open question. Furthermore, the optimal hyperparameter for the high-entropy token ratio, while robust in our tests, may require further tuning for new tasks and model architectures. Future work should explore these avenues to validate and refine the HES framework.

| Method | AIME24 | AIME25 | HMMT23 | HMMT24 | HMMT25 | Oly(E) | Oly(H) | GPQA | AVG |
|---|---|---|---|---|---|---|---|---|---|
| *Per-Query Selection ($k = 4$)* | | | | | | | | | |
| Random | 17.08 | **22.50** | 10.63 | **12.71** | 9.17 | 12.38 | 2.44 | 27.81 | 14.34 |
| Highest-HES | **18.54** | 19.38 | **14.79** | 10.83 | **11.88** | **14.31** | 2.88 | **28.13** | **15.09** |
| Lowest-HES | 14.58 | 18.13 | 12.08 | 10.21 | 9.38 | 12.06 | 2.06 | 27.30 | 13.23 |
| *Global Pool Selection ($k = 4$)* | | | | | | | | | |
| Random | 14.17 | 19.58 | 11.04 | **11.46** | 10.56 | 12.81 | **3.35** | 27.05 | 13.75 |
| Highest-HES | **17.92** | **20.63** | **14.38** | 11.25 | **12.29** | **15.19** | 3.13 | 25.09 | **14.99** |
| Lowest-HES | 1.46 | 10.00 | 5.21 | 5.21 | 2.92 | 3.56 | 1.31 | **27.43** | 7.14 |

Table 7: Performance comparison of RFT strategies using DeepSeek-R1-Distilled-Llama-8B.

| Method | Ratio | AIME24 | AIME25 | HMMT23 | HMMT24 | HMMT25 | Oly(E) | Oly(H) | GPQA | AVG |
|---|---|---|---|---|---|---|---|---|---|---|
| Full-Dataset | 100 | **48.96** | 32.50 | 31.67 | 25.83 | 22.50 | 38.38 | 5.81 | 37.00 | 30.33 |
| Random | 20 | 30.00 | 30.21 | 24.17 | 19.79 | 17.50 | 25.00 | 5.56 | 41.73 | 24.25 |
| Highest-AvgE | 20 | 39.38 | 26.67 | 22.29 | 19.38 | 18.75 | 26.38 | 4.38 | 43.28 | 25.06 |
| Highest-AvgHE | 20 | 39.58 | 29.58 | 23.54 | 18.75 | 20.00 | 29.13 | 3.63 | 44.19 | 26.05 |
| Lowest-HES | 20 | 15.42 | 12.29 | 6.04 | 8.13 | 6.04 | 5.63 | 2.13 | 35.54 | 11.40 |
| Highest-HES | 20 | 48.54 | 32.92 | 30.63 | 23.33 | 22.29 | 37.94 | **6.56** | **49.78** | 31.50 |
| Highest-HES | 80 | **48.96** | **35.63** | **36.04** | **26.67** | **22.71** | **41.88** | 6.00 | 40.37 | **32.28** |

Table 8: Performance comparison of SFT using Qwen3-8B-Base on OpenR1-Math-220k.

# D  DATASET

**OpenR1-Math-220k**    is a massive dataset designed to advance mathematical reasoning in LLMs. It comprises 220,000 math problems sourced from the NuminaMath 1.5 collection. A key feature of this dataset is that each problem is accompanied by two to four distinct reasoning traces, or step-by-step solutions, generated by the DeepSeek R1 model. To ensure reliability, these traces underwent a rigorous verification process: most were validated using Math Verify, while 12% were assessed by a Llama-3.3-70B-Instruct judge. Critically, every problem in the dataset is guaranteed to contain at least one reasoning trace that leads to a correct answer. The dataset is organized into two distinct splits: (1) The default split, containing 94,000 problems, is the recommended version for training. It has been shown to yield the best performance improvements after SFT. (2) The extended split expands the collection to 131,000 problems by incorporating additional data sources. While this provides a greater volume of reasoning traces, models fine-tuned on this split have shown slightly lower performance, which is likely because the questions from the added sources are less difficult than those in the core dataset.

**Open-Math-Reasoning**    is also a large-scale dataset created to train LLMs in advanced mathematical reasoning. At its core, it contains 306,000 unique mathematical problems sourced from the Art of Problem Solving (AoPS) forums. These are complemented by a massive volume of solutions, including 3.2 million long-form Chain-of-Thought (CoT) examples and 1.7 million Tool-Integrated Reasoning (TIR) solutions, showcasing diverse and detailed problem-solving paths. The dataset also includes 566,000 "GenSelect" samples, which are designed to teach models how to evaluate and select the most promising solution from multiple candidates. To further expand its utility, an additional 193,000 problems from the AoPS forums are included without solutions. The creation process involved state-of-the-art models, with Qwen2.5-32B-Instruct used for preprocessing problems and both DeepSeek-R1 and QwQ-32B for generating the high-quality solutions. Notably, this dataset was a foundational component of the winning submission to the AIMO-2 Kaggle competition, highlighting its power and effectiveness in building top-tier AI reasoning systems.

**Deepscaler**    is a curated collection of approximately 40,000 unique mathematics problem-answer pairs, specifically assembled to train advanced problem-solving models. This comprehensive dataset is compiled from a variety of high-quality and challenging sources. It features a deep historical archive of problems from prestigious competitions, including the American Invitational Mathematics Examination (AIME) from 1984 to 2023 and the American Mathematics Competition (AMC)

| Method | Ratio | AIME24 | AIME25 | HMMT23 | HMMT24 | HMMT25 | Oly(E) | Oly(H) | GPQA | AVG |
|---|---|---|---|---|---|---|---|---|---|---|
| Full-Dataset | 100 | 39.17 | 31.04 | 29.58 | 24.17 | 20.83 | 37.00 | 4.63 | 33.96 | 27.55 |
| Random | 20 | 30.83 | 26.04 | 20.42 | 16.67 | 17.08 | 20.06 | 3.25 | 38.19 | 21.57 |
| Highest-AvgE | 20 | 34.58 | 28.13 | 21.46 | 14.79 | 17.92 | 23.63 | 3.06 | 36.56 | 22.52 |
| Highest-AvgHE | 20 | 33.75 | 25.42 | 21.67 | 17.29 | 15.83 | 23.56 | 2.19 | 39.24 | 22.37 |
| Lowest-HES | 20 | 16.04 | 16.46 | 9.17 | 7.08 | 6.46 | 7.00 | 2.13 | 36.90 | 12.66 |
| Highest-HES | 20 | 38.13 | 31.88 | 22.71 | 20.42 | 18.54 | 28.00 | 4.88 | **39.39** | 25.49 |
| Highest-HES | 80 | **46.25** | **33.96** | **31.88** | **25.83** | **21.46** | **39.44** | **5.13** | 34.85 | **29.85** |

Table 9: Performance comparison of SFT using Qwen3-4B-Base on Open-Math-Reasoning.

from years prior to 2023. To ensure breadth and diversity, the collection is further supplemented with content from the Omni-MATH and Still datasets.

# E SUPPLEMENTARY RESULTS

To further validate the robustness and generalizability of the HES metric, we conducted additional experiments across different datasets, model sizes, and architectures. This section presents detailed results that complement the main findings.

## E.1 CONSISTENCY ACROSS DATASETS AND MODEL SCALES

We replicated the SFT experiments on the OpenR1-Math-220k dataset using the Qwen3-8B-Base model (Table 8) and on the Open-Math-Reasoning dataset using the smaller Qwen3-4B-Base model (Table 9).

The results consistently demonstrate the efficacy of HES-based data selection:

- **Superiority of Highest-HES:** In both settings, training on the top 20% of data ranked by HES (Highest-HES) achieves performance comparable to or better than random selection, often approaching the full-dataset baseline. Notably, on OpenR1-Math-220k, the top 20% HES subset outperforms the full dataset on the GPQA benchmark (49.78% vs. 37.00%).

- **Effective Noise Filtering:** Training on the top 80% of data (i.e., pruning the bottom 20% with the lowest HES) consistently yields the best overall performance. For instance, on OpenR1-Math-220k, the Highest-HES (80) strategy achieves an average accuracy of 32.28%, surpassing the full-dataset performance of 30.33%. Similarly, on Open-Math-Reasoning with the 4B model, the top 80% subset achieves an average of 29.85%, outperforming the full dataset's 27.55%.

- **Detrimental Effect of Low-HES Data:** Consistent with our main findings, training on the bottom 20% of data ranked by HES (Lowest-HES) leads to a catastrophic drop in performance, significantly underperforming even the random baseline. This confirms that HES effectively identifies low-quality data that can be harmful to model training.

## E.2 ROBUSTNESS ACROSS MODEL ARCHITECTURES

To ensure that the effectiveness of HES is not limited to the Qwen model family, we conducted RFT experiments using the DeepSeek-R1-Distilled-Llama-8B model (Table 7). We compared random selection, HES-based selection, and inverse HES selection across both per-query and global pool settings with $k = 4$.

The results on the Llama-based model align with our previous findings:

- **HES Outperforms Random Selection:** In both per-query and global pool settings, selecting the highest HES responses consistently yields better average performance than random selection. For example, in the per-query setting, `Highest-HES` achieves an average of 15.09%, compared to 14.34% for random selection.

- **Criticality of Global Pool Selection:** The discriminative power of HES is particularly evident in the global pool setting. While random selection achieves 13.75%, `Highest-HES`

improves this to 14.99%. Conversely, selecting the lowest HES responses results in a severe performance degradation to 7.14%, highlighting the metric's ability to distinguish high-quality reasoning paths from poor ones across different model architectures.

These supplementary results reinforce the conclusion that HES is a robust, model-agnostic metric for data selection in LLM reasoning, capable of improving performance across various datasets, model scales, and architectures.

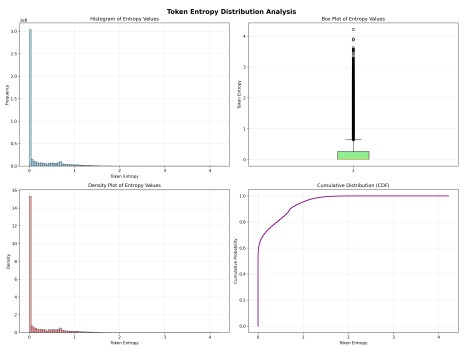 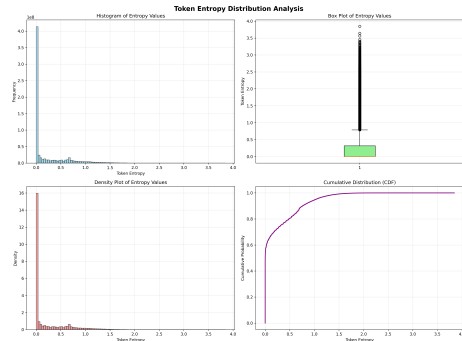

(a) Token entropy distribution of OpenR1-Math-220k.

(b) Token entropy distribution of Open-Math-Reasoning.

Figure 5: Wordcloud of high-entropy tokens.

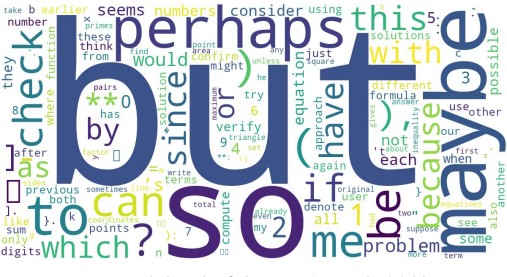

(a) Wordcloud of OpenR1-Math-220k.

(b) Wordcloud of Open-Math-Reasoning.

Figure 6: Wordcloud of high-entropy tokens.

## F  ANALYSIS

### F.1  DISCRIMINATIVE ABILITY OF HES

To validate the effectiveness of HES as a quality metric, we analyze its ability to discriminate between high- and low-quality samples against several alternative entropy-based metrics. This analysis is conducted across multiple benchmarks, including AIME24, HMMT23, HMMT24, and HMMT25. As illustrated by the score distributions in Figure 8 to Figure 11, the HES demonstrates a significantly clearer separation between correct and incorrect reasoning paths compared to other metrics. Specifically, the distributions of HES scores for correct and incorrect samples show minimal overlap. The mean HES for incorrect paths is substantially higher than for correct paths, indicating that flawed reasoning corresponds to a higher cumulative sum of uncertainty at pivotal moments. In contrast, global metrics like the AveE and even the more targeted AvgHET show less separation between their means for correct and incorrect samples. This analysis confirms that the cumulative signal captured by HES is a more robust and reliable indicator of reasoning quality than metrics based on either global averages or the average intensity of pivotal moments alone. The clear separation validates its use as a foundational metric for our data selection framework.

Additionally, Figure 7 provides specific entropy value references.

## F.2 HIGH-ENTROPY TOKEN DISTRIBUTION

To better understand the characteristics of our training data, we conduct a detailed analysis of the token-level entropy distributions for both the OpenR1-Math-220k and Open-Math-Reasoning datasets. The analysis, illustrated in Figure 5a and 5, reveals that the entropy distribution is heavily right-skewed. The vast majority of tokens exhibit very low entropy, corresponding to routine or predictable steps, while a small minority of tokens accounts for the high-entropy tail of the distribution. Our calculations show that the entropy value corresponding to the top 0.5% of high-entropy tokens is approximately 1.55 for OpenR1-Math-220k and 1.60 for Open-Math-Reasoning. This data-driven finding provides an empirical basis for our threshold choice of $1.6$ in the $HES_{absolute}$ experiments. Furthermore, a qualitative analysis using word clouds shown in Figure 6 confirms that these high-entropy tokens are not random noise; they typically correspond to semantically significant forking points that guide the reasoning process. Taken together, these findings validate our core premise: a small, identifiable subset of tokens represents the most complex decision points, motivating our focus on the HES metric.

## F.3 TOKEN DISTRIBUTION EXAMPLE

The entropy distribution of a sample is shown as follows. We can see that high entropy words are small in number of forking points, while low entropy words are large in number.

# G COMPUTATIONAL EFFICIENCY AND COST ANALYSIS

A primary concern in data selection is whether the cost of computing the metric outweighs the benefits of reduced training data. In this section, we analyze the computational overhead of HES in both online generation (RFT/RL) and offline screening (SFT) scenarios, demonstrating that HES is a highly cost-effective metric.

## G.1 ONLINE SETTINGS (RFT AND RL): ZERO EXTRA INFERENCE OVERHEAD

In RFT and RL, the model generates reasoning trajectories (rollouts) in real-time. In these scenarios, calculating HES incurs **zero extra inference overhead** in terms of model forward passes.

- **Piggybacking on Generation:** During the auto-regressive generation process, the model must perform a forward pass at every time step $t$ to compute the probability distribution over the vocabulary, $P(x_t|x_{<t})$, in order to sample the next token.

- **Synchronous Calculation:** The entropy calculation is performed synchronously using these intermediate logits. We simply calculate the entropy of the sampled token's distribution and accumulate it in memory. This operation involves only negligible vector operations (complexity $O(V)$, where $V$ is vocabulary size), which is orders of magnitude faster than the matrix multiplications required for the model's forward pass.

- **No Additional Forward Pass:** Unlike methods that require a separate reward model or a second pass over the completed sequence, HES reuses the computation already performed for generation. Thus, it does not increase the inference latency or compute budget during rollout.

## G.2 OFFLINE SETTINGS (SFT): EFFICIENT SCREENING VIA PROXY MODELS

For SFT, data selection is typically performed offline on static datasets. While calculating HES requires a forward pass to obtain logits, we demonstrate that this cost can be minimized by using smaller **Proxy Models**.

**Small Model Transferability:** We investigated whether HES scores derived from smaller models correlate with data quality for larger models. As shown in Table 1, we used a Qwen3-0.6B model to screen data for training a Qwen3-8B model.

- **Performance Consistency:** The 8B model trained on data selected by the 0.6B proxy achieved an average accuracy of 32.12%, comparable to (and even slightly exceeding) the performance when using the 8B model for selection (31.14%).
- **Cost Reduction:** The inference cost of a 0.6B model is approximately $1/13$th that of an 8B model. By using a small proxy model, we can screen massive datasets at a fraction of the cost.

### G.3 OVERALL RETURN ON INVESTMENT (ROI)

The computational cost of HES selection should be weighed against the savings in training compute.

- **Training vs. Inference:** The computational cost of training (backward pass + optimizer) is significantly higher than inference (forward pass only).
- **Data Pruning:** Our experiments show that training on the top 20% of HES-selected data matches or exceeds full-dataset performance. This represents an **80% reduction** in total training FLOPs.

In summary, the inference cost of calculating HES (especially when using small proxy models) is negligible compared to the massive savings achieved by avoiding gradient updates on 80% of low-value data. This makes HES a highly efficient strategy for large-scale LLM training.

<think>\nOkay, let me try to figure out how to evaluate this expression: 0 - 1 - 2 + 3 - 4 + 5 + 6 + 7 - 8 + ... + 2000. The problem says that the terms with minus signs are exactly the powers of two. Hmm, so first, I need to make sure I understand the pattern here. Let me list out the terms step by step to see if I can spot the pattern.\n\nStarting from 0:\n\nTerm 0: 0 (but the first term is 0, then subtract 1, subtract 2, add 3, subtract 4, etc.)\n\nWait, let's write it out more clearly. The expression is:\n\n0 - 1 - 2 + 3 - 4 + 5 + 6 + 7 - 8 + ... + 2000\n\nSo, starting from 0, then subtract 1, subtract 2, add 3, subtract 4, add 5, add 6, add 7, subtract 8, and so on. The minus signs are on powers of two. Let's check:\n\nPowers of two are 1, 2, 4, 8, 16, 32, ..., up to 1024 since $2^{10} = 1024$, and next would be 2048 which is beyond 2000. So all the powers of two from 1 up to 1024 are subtracted, and all other numbers are added. Let me verify this with the given expression.\n\nStarting from term 0:\n\n0 (term 0) is separate. Then:\n\n-1 (term 1), which is $2^0 =1$\n\n-2 (term 2), which is $2^1 =2$\n\n+3 (term 3)\n\n-4 (term 4), which is $2^2=4$\n\n+5 (term5), +6 (term6), +7 (term7)\n\n-8 (term8), which is $2^3=8$\n\nThen from 9 to 15: all added?\n\nWait, next power of two is 16, so terms 9 to 15 would be added, then term16 is subtracted. So the pattern is that whenever the term number is a power of two, it's subtracted; otherwise, it's added. But wait, the problem says "the terms with minus signs are exactly the powers of two." So that's correct. So each term in the series is being added except for the terms that are powers of two, which are subtracted.\n\nBut note that the first term is 0. So the expression is:\n\n0 (term0) -1 (term1) -2 (term2) +3 (term3) -4 (term4) +5 (term5) +6 (term6) +7 (term7) -8 (term8) +... +2000.\n\nSo the series starts at 0, then for each term from 1 to 2000, if the term is a power of two, it's subtracted; otherwise, it's added.\n\nWait, but does the problem say "the terms with minus signs are exactly the powers of two"? So, all the subtracted terms are powers of two, and all powers of two are subtracted. So, starting from 1 to 2000, every number that is a power of two is subtracted, and all others are added. The initial 0 is just there. So, the entire expression can be written as:\n\n$0 + (sum from n=1 to 2000 of (-1)^{n is power of two} * n )$\n\nBut how do we compute that? Let's think.\n\nSo, the total sum would be 0 plus the sum from n=1 to 2000 of n, except that whenever n is a power of two, we subtract 2n instead of adding n. Wait, no. Wait, if a term is subtracted instead of added, the difference is that instead of +n, we have -n. So the total sum would be (sum from n=1 to 2000 of n) - 2*(sum of all powers of two from 1 to 2000).\n\nBecause for each power of two term, we subtract it instead of adding it, so we have to subtract twice that term. Because normally it's +n, but now it's -n, so the difference is -2n. Therefore, total sum is $sum_{n=1}^{2000} n - 2*sum_{k=0}^{m} 2^k$, where $2^m <=2000$.\n\nWait, let's verify that. Let's say $S = sum_{n=1}^{2000} n$, which is (2000)(2001)/2. Then, the terms which are powers of two are subtracted instead of added. So each power of two term is contributing -n instead of +n, so the difference is -2n. Therefore, the total sum would be S - 2*sum_{powers of two <=2000} n. That seems correct.\n\nSo first, compute $S = sum$ from 1 to 2000: S = (2000)(2001)/2.\n\nThen, compute T = sum of all powers of two up to 2000.\n\nWhat's the sum of powers of two from 1 to 2000?\n\nPowers of two are 1, 2, 4, 8, ..., 1024. Since $2^{10} = 1024$, $2^{11} = 2048$ which is over 2000. So the exponents go from 0 to 10, so the sum is $2^{(10+1)} -1 = 2048 -1 = 2047$. Wait, but wait, the sum of $2^0 + 2^1 + ... +2^m = 2^{(m+1)} -1$. So here, the powers of two up to 1024 (which is $2^{10}$) would be $2^0 + 2^1 + ... +2^{10} = 2^{11} -1 = 2048 -1 = 2047$. Therefore, T = 2047.\n\nTherefore, the total sum would be S - 2*T = (2000*2001/2) - 2*2047.\n\nCompute that:\n\nFirst, compute S = 2000*2001/2. Let's calculate that:\n\n2000*2001 = 2000*(2000 +1) = 2000^2 + 2000 = 4,000,000 + 2,000 = 4,002,000\n\nThen, divide by 2: 4,002,000 /2 = 2,001,000.\n\nThen, 2*T = 2*2047 = 4094.\n\nTherefore, total sum is 2,001,000 - 4,094 = let's compute that.\n\n2,001,000 - 4,000 = 1,997,000\n\nThen subtract 94: 1,997,000 - 94 = 1,996,906.\n\nWait, wait. Wait 2,001,000 - 4,094:\n\n2,001,000 - 4,000 = 1,997,000\n\nThen subtract

94 more: 1,997,000 -94 = 1,996,906.\n\nYes. So the total sum is 1,996,906.\n\nWait, but hold on. The first term in the given expression is 0. So in the problem, the expression starts at 0. So when we calculated sum from n=1 to 2000, we included all terms starting from 1. So the initial 0 is separate. Wait, but the problem is written as:\n\n0 -1 -2 +3 -4 +5 +6 +7 -8 + ... +2000\n\nSo starting from 0, then all terms from 1 to 2000 are included with the signs. Therefore, in our calculation above, S is sum from 1 to 2000, and then we subtract 2*T. Then, since the expression is 0 + [sum from 1 to 2000 with signs], so the total is indeed 0 + (S - 2*T). Therefore, our calculation is correct.\n\nBut let me verify this with a small example. Let's take a small N, say up to 8.\n\nTest case: Compute 0 -1 -2 +3 -4 +5 +6 +7 -8.\n\nAccording to our formula:\n\nsum from 1 to 8: 8*9/2 = 36\n\nsum of powers of two up to 8: 1 +2 +4 +8=15\n\nTotal sum: 0 + (36 - 2*15) = 0 + (36 -30) =6\n\nBut let's compute manually:\n\n0 -1 -2 +3 -4 +5 +6 +7 -8\n\nCompute step by step:\n\n0 -1 = -1\n\n-1 -2 = -3\n\n-3 +3 = 0\n\n0 -4 = -4\n\n-4 +5 = 1\n\n1 +6 =7\n\n7 +7=14\n\n14 -8=6\n\nWhich matches the formula's result. Therefore, the formula seems correct.\n\nAnother test case: up to 16. Let's compute using formula and manually.\n\nFormula:\n\nsum from 1 to16:16*17/2=136\n\nsum of powers of two up to 16:1+2+4+8+16=31\n\nTotal sum:136 -2*31=136-62=74\n\nManually:\n\n0 -1 -2 +3 -4 +5 +6 +7 -8 +9 +10 +11 +12 +13 +14 +15 -16\n\nLet's compute step by step:\n\n0 -1 = -1\n\n-1 -2 = -3\n\n-3 +3 =0\n\n0 -4 = -4\n\n-4 +5=1\n\n1 +6=7\n\n7 +7=14\n\n14 -8=6\n\n6 +9=15\n\n15 +10=25\n\n25 +11=36\n\n36 +12=48\n\n48 +13=61\n\n61 +14=75\n\n75 +15=90\n\n90 -16=74\n\nWhich matches the formula. Therefore, the formula works.\n\nTherefore, for the original problem, the answer should be 1,996,906.\n\nBut let me check again the sum of powers of two up to 2000. The powers of two are 1, 2, 4, 8, 16, 32, 64, 128, 256, 512, 1024. Let's check:\n\n2^0 =1\n\n2^1=2\n\n2^2=4\n\n2^3=8\n\n2^4=16\n\n2^5=32\n\n2^6=64\n\n2^7=128\n\n2^8=256\n\n2^9=512\n\n2^10=1024\n\n2^11=2048 >2000, so stop here.\n\nSo sum is 1+2+4+8+16+32+64+128+256+512+1024. Let's compute that:\n\n1+2=3\n\n3+4=7\n\n7+8=15\n\n15+16=31\n\n31+32=63\n\n63+64=127\n\n127+128=255\n\n255+256=511\n\n511+512=1023\n\n1023+1024=2047\n\nYes, so sum of powers of two up to 2000 is 2047. Therefore, the calculation is correct.\n\nSo total sum is 2,001,000 -4,094=1,996,906.\n\nTherefore, the answer is \boxed{1996906}.\n\n**Final Answer**\n\boxed{1996906}\n</think>\n\nTo evaluate the expression \(0 - 1 - 2 + 3 - 4 + 5 + 6 + 7 - 8 + \ldots + 2000\), where the terms with minus signs are exactly the powers of two, we can follow these steps:\n\n1. **Identify the sum of all terms from 1 to 2000**:\n  \[\n  S = \sum_{n=1}^{2000} n = \frac{2000 \times 2001}{2} = 2001000\n  \]\n\n2. **Identify the sum of all powers of two up to 2000**:\n  The powers of two up to 2000 are \(1, 2, 4, 8, 16, 32, 64, 128, 256, 512, 1024\). The sum of these powers of two is:\n  \[\n  T = 1 + 2 + 4 + 8 + 16 + 32 + 64 + 128 + 256 + 512 + 1024 = 2047\n  \]\n\n3. **Adjust the total sum by subtracting twice the sum of the powers of two**:\n  Since each power of two is subtracted instead of added, the total adjustment is \(2 \times T\). Therefore, the total sum is:\n  \[\n  \text{Total Sum} = S - 2T = 2001000 - 2 \times 2047\n  \]\n\n4. **Calculate the final result**:\n  \[\n  2001000 - 4094 = 1996906\n  \]\n\nThus, the evaluated value of the given expression is \(\boxed{1996906}\)

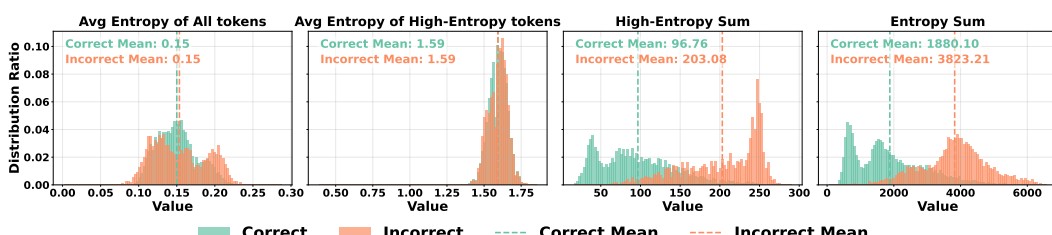

Figure 7: Comparative analysis of discriminative ability between HES and other metrics based on 512 responses per problem sampled by Qwen3-14B on AIME 2025. The x-axis is specific entropy value.

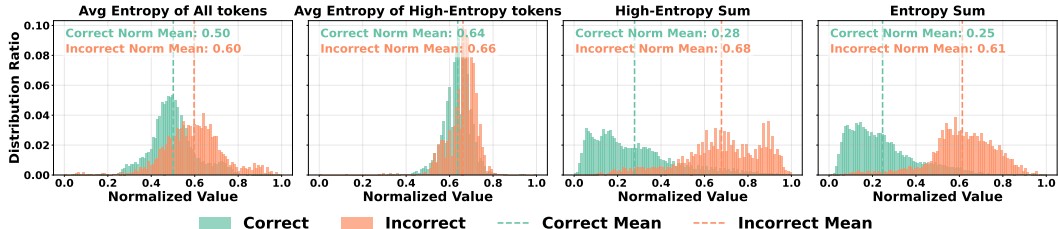

Figure 8: Comparative analysis of discriminative ability between HES and other metrics based on 512 responses per problem sampled by Qwen3-14B on AIME 2024.

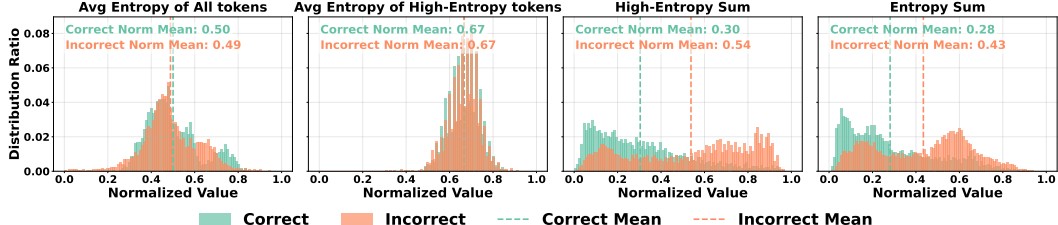

Figure 9: Comparative analysis of discriminative ability between HES and other metrics based on 512 responses per problem sampled by Qwen3-14B on HMMT 2023.

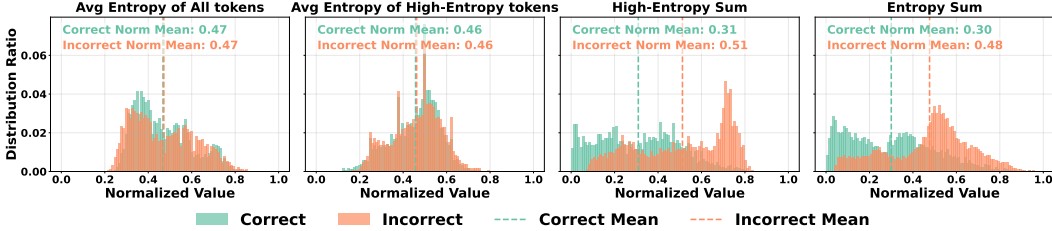

Figure 10: Comparative analysis of discriminative ability between HES and other metrics based on 512 responses per problem sampled by Qwen3-14B on HMMT 2024.

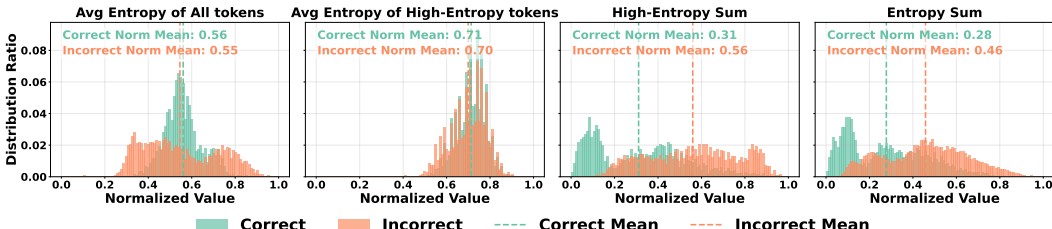

Figure 11: Comparative analysis of discriminative ability between HES and other metrics based on 512 responses per problem sampled by Qwen3-14B on HMMT 2025.

