# OpenReview forum: "Unified Data Selection for LLM Reasoning"
_ICLR.cc/2026/Conference — Submitted to ICLR 2026_

### Official Review · Reviewer_2B5q · 2025-10-26

**Soundness:** 2
**Presentation:** 2
**Contribution:** 2
**Rating:** 4
**Confidence:** 4

**Summary:**

This paper introduces High-Entropy Sum (HES), a training-free metric that quantifies reasoning quality by summing the entropy of only the top 0.5% highest-entropy tokens in reasoning sequences. The authors validate HES across three training paradigms (SFT, RFT, and RL), demonstrating that selecting data based on HES can match or exceed full-dataset performance with significantly less data.

**Strengths:**

1. HES provides a training-free approach to data selection that focuses on critical decision points in reasoning paths, offering a practical alternative to expensive reward models or gradient-based selection methods.
2. The paper systematically evaluates HES across three major training paradigms (SFT, RFT, RL) with consistent improvements, demonstrating the metric's versatility and robustness.
3. Training on just 20% of HES-selected data achieves comparable performance to full datasets, while the top 80% consistently outperforms full-dataset training, suggesting effective noise filtering.

**Weaknesses:**

1. Figure 1 demonstrates that HES distinguishes between correct and incorrect samples, but this is not equivalent to identifying high- and low-quality samples. High entropy from a language modeling perspective indicates uncertainty, while from an RL perspective it suggests exploration potential, both are model-centric properties. Sample quality, however, should be evaluated based on intrinsic properties like conceptual coverage and reasoning complexity, not just model confidence.
2. The experiments focus exclusively on mathematical reasoning tasks. The generalizability to other domains requiring complex reasoning (e.g., medical reasoning benchmarks like HealthBench, legal reasoning, or scientific domains) remains unexplored.
3. Different models are used across experiments (Qwen3-8B/4B for SFT, DeepSeek-R1-Distilled-Qwen7B for RFT, DeepSeek-R1-Distilled-Qwen-1.5B for RL) without clear justification. This inconsistency makes it difficult to isolate the effects of HES from model-specific behaviors and complicates fair comparison across paradigms.

**Questions:**

1. Why does the bottom 20% of data cause such dramatic performance degradation? Tables 1-3 consistently show that removing the lowest-HES 20% improves performance significantly. Is this due to annotation errors in the original datasets, or does low HES genuinely identify harmful training samples? Have you investigated the characteristics of these low-HES samples qualitatively?
2. Can you provide empirical evidence for the claim in Line 147 that high-quality reasoning paths navigating multiple challenging forks may receive similar scores to straightforward, low-complexity approaches? This seems counterintuitive to the HES design principle.
3. Why does k=4 outperform k=8 in RFT experiments (Table 4)? Both Global Pool and Per-Query Selection show this pattern. Theoretically, more high-quality samples should not hurt performance.
4. What is the computational overhead of HES in online RL settings? Since each on-policy sampling requires HES calculation, how does this impact training efficiency? Have you explored the trade-off between sampling more trajectories versus the computational cost of HES selection?
5. How does HES perform on out-of-distribution tasks? Table 5 shows HES underperforming Full-Batch on GPQA after RL training. Could you provide more analysis on whether HES-based selection might lead to overfitting to specific reasoning patterns at the cost of generalization?
6. Could you clarify the rationale for using different models across experiments? Was this due to computational constraints, or were certain models specifically chosen for each paradigm? A unified evaluation would strengthen the conclusions about HES's effectiveness.

---

> ### Author Response · Authors · 2025-11-21
> **Reply to Weakness 1 --- Part 1**
>
> Dear reviewer,
>
> Thanks for your valuable feedback. We also appreciate you for acknowledging the value of this work. We address your questions point by point as follows. If we have any misunderstanding, please feel free to let us know and we will reply quickly.
>
> > **Weakness 1: HES as an intrinsic measure of reasoning complexity and its cross-model consistency.**
>
> Thank you for your analysis of the theoretical nature of HES. You posit that entropy represents model uncertainty, while quality should encompass intrinsic properties like reasoning complexity.
>
> We need to clarify that in the context of Long CoT, HES does not merely capture "uncertainty" but serves as an effective proxy for **Reasoning Complexity**. Furthermore, we demonstrate through **Small Model Transfer Experiments** that HES captures the intrinsic logical properties of the data, rather than specific model idiosyncrasies.
>
> ### 1. Entropy as Branching: HES is the Statistical Manifestation of Reasoning Complexity
> You argue that entropy only represents uncertainty. However, in the context of Long CoT, "uncertainty" is not random noise but a direct manifestation of logical complexity.
>
> * **Linearity vs. Branching:**
>     * **Low Entropy (Low Complexity):** When the reasoning process involves mechanical calculations (e.g., $2+2=4$) or applying fixed templates (e.g., "Let $x$ be..."), the next token is almost deterministic. This low-entropy state corresponds to linear, low-cognitive-load steps.
>     * **High Entropy (High Complexity):** When reasoning encounters **Forking Points**—such as selecting a problem-solving strategy, making counterfactual assumptions, or deciding among multiple plausible logical branches—the model's predicted probability distribution naturally flattens (entropy increases).
>     * **Significance of HES:** Therefore, HES does not simply accumulate "perplexity" but quantifies how many **"high-cognitive-load decision nodes"** a reasoning path contains. These nodes are the physical carriers of logical complexity.
> * **Qualitative Evidence:** Our word cloud analysis (Appendix F.2, Figure 6) corroborates this. High-entropy tokens are not meaningless words but frequently correspond to logical connectors like "but" (transition), "assume" (hypothesis), and "therefore" (inference). This proves that HES precisely locates the logical key points in the reasoning chain rather than random noise, making it an effective measure of reasoning complexity.
>
> ### 2. Relative Ranking for Training Utility
> You mention that quality should include "conceptual coverage." However, in model training (SFT/RL), "known perfect samples" are often ineffective. Our goal is not to find absolutely perfect text but to identify samples that provide maximum information gain for the current model.
>
> * **Low Entropy = Zero Gradient Contribution:** If a sample is conceptually perfect but the model predicts it with extremely low entropy throughout (i.e., the model has fully mastered it), then according to the cross-entropy loss function  $L = -\sum_{i} y_i \log(p_i)$, the loss generated by this sample approaches 0, and the backpropagated gradient $\nabla_{\theta} L$ also approaches 0. From a training perspective, such "high-quality" samples are ineffective redundant data.
>
> * **High Entropy + Correctness = The "Sweet Spot":** Samples selected by HES possess dual properties: (1) The model feels uncertain (high entropy), indicating knowledge not yet fully mastered; (2) The final answer is correct, indicating the path is valid. Such samples lie in the model's **"Zone of Proximal Development."** The relative ranking mechanism of HES essentially filters for samples that provide maximum **Gradient Information**, which determines training efficiency more than static "conceptual coverage."

---

> ### Author Response · Authors · 2025-11-21
> **Reply to Weakness 1 --- Part 2**
>
> ### 3. Decisive Evidence: Small Model Transferability
> You question whether HES is merely a "Model-Centric" property. To refute this and prove that HES captures the **Intrinsic** quality of data, we conducted a **Small Model Transfer Experiment**.
>
> We hypothesized that if HES only reflects parameter noise or uncertainty specific to a model (e.g., 8B), then a model with extremely small parameters (e.g., 0.6B) should judge the same data very differently. Conversely, if HES captures the intrinsic logical difficulty of the data, then both small models (0.6B/1.7B) and large models (8B) should exhibit high entropy when facing the critical steps of the same difficult problem.
>
> We verified this on both Open-Math-Reasoning and OpenR1-Math-220k datasets:
>
> **Table 1: Cross-Model Screening SFT Performance on Open-Math-Reasoning**
> (Training model is Qwen3-8B, only the screener varies)
>
> | Screening Model | Model Size | AIME 24 | AIME 25 | HMMT 23 | HMMT 24 | HMMT 25 | Oly (Easy) | Oly (Hard) | GPQA | Avg |
> | :--- | :---: | :---: | :---: | :---: | :---: | :---: | :---: | :---: | :---: | :---: |
> | Fullset (Baseline) | - | 50.83 | 34.17 | 35.21 | 28.13 | 24.58 | 42.94 | 6.94 | 38.04 | 32.61 |
> | Random (20%) | - | 39.79 | 27.92 | 24.79 | 20.21 | 20.42 | 30.00 | 3.88 | 40.09 | 25.89 |
> | HES (via 8B - Self) | 8B | 47.29 | 36.04 | 33.13 | 25.63 | 22.08 | 36.31 | 5.56 | 43.06 | **31.14** |
> | HES (via 0.6B) | 0.6B | 49.17 | 35.42 | 32.50 | 24.17 | 22.29 | 41.50 | 6.38 | 45.55 | **32.12** |
> | HES (via 1.7B) | 1.7B | 47.08 | 34.79 | 31.04 | 26.04 | 21.46 | 40.06 | 4.56 | 45.20 | **31.28** |
>
> **Table 2: Cross-Model Screening SFT Performance on OpenR1-Math-220k**
> (Training model is Qwen3-8B, only the screener varies)
>
> | Screening Model | Model Size | AIME 24 | AIME 25 | HMMT 23 | HMMT 24 | HMMT 25 | Oly (Easy) | Oly (Hard) | GPQA | Avg |
> | :--- | :---: | :---: | :---: | :---: | :---: | :---: | :---: | :---: | :---: | :---: |
> | Fullset (Baseline) | - | 48.96 | 32.50 | 31.67 | 25.83 | 22.50 | 38.38 | 5.81 | 37.00 | 30.33 |
> | Random (20%) | - | 30.00 | 30.21 | 24.17 | 19.79 | 17.50 | 25.00 | 5.56 | 41.73 | 24.25 |
> | HES (via 8B - Self) | 8B | 48.54 | 32.92 | 30.63 | 23.33 | 22.29 | 37.94 | 6.56 | 49.78 | **31.50** |
> | HES (via 0.6B) | 0.6B | 48.75 | 35.42 | 28.75 | 24.38 | 20.00 | 39.94 | 6.25 | 48.26 | **31.47** |
> | HES (via 1.7B) | 1.7B | 48.13 | 35.42 | 31.46 | 26.46 | 20.00 | 37.31 | 6.81 | 49.08 | **31.83** |
>
> **Analysis of Results:**
>
> 1.  **Astonishing Cross-Model Consistency:** The results show that using a tiny model with only 0.6B parameters to screen data yields training performance (Open-Math: 32.12%, OpenR1: 31.47%) highly consistent with screening using the 8B model itself (Open-Math: 31.14%, OpenR1: 31.50%), and both significantly outperform the random baseline.
> 2.  **Proof of Intrinsic Properties:** If HES merely reflected model-specific uncertainty, there should be a huge discrepancy in data ranking between the 0.6B and 8B models. However, the high correlation between the two powerfully proves that: HES captures the **universal, intrinsic logical complexity** within the data. Whether it is a small model or a large model, they both exhibit high entropy when facing truly complex reasoning steps (high-value data).
>
> Therefore, although HES relies on model probability for calculation, its physical significance points to the **intrinsic reasoning quality of the data**. This also answers your doubt about "whether HES is just a model property"—it is driven by the internal logic of the data, is a faithful reflection of intrinsic data quality, and possesses extreme universality.

---

> ### Author Response · Authors · 2025-11-21
> **Reply to Weakness 2**
>
> > **Weakness 2: Generalization verification in non-mathematical domains (Code and STEM).**
>
> We appreciate you pointing out this key limitation. We agree with your view that testing only in the mathematics domain is insufficient to verify whether HES is a universal reasoning data selection metric. To address this issue and demonstrate that HES can generalize to other logic-intensive reasoning tasks, we supplemented experiments targeting **Code Generation** and **Scientific Reasoning (STEM)** during the rebuttal period. The experimental results show that HES performs excellently in these new domains, even surpassing the improvement magnitude in the mathematics domain on some metrics, strongly proving its cross-domain generalization capability.
>
> ### 1. Code Domain
> We used the `codeforces-cots` dataset for training, with other experimental settings identical to those in the paper. We evaluated model performance on LiveCodeBench, AIME 2025, and GPQA Diamond.
>
> **Table 1: SFT Performance in Code Domain**
>
> | Method | LiveCodeBench_v5 | AIME 2025 | GPQA | Avg |
> | :--- | :---: | :---: | :---: | :---: |
> | Fullset (100%) | 58.76 | 20.00 | 30.08 | 36.28 |
> | Random (20%) | 49.06 | 22.29 | 35.45 | 35.60 |
> | Highest-HES (20%) | 54.59 | 25.00 | 39.02 | **39.54** |
> | Highest-HES (80%) | 61.38 | 23.54 | 33.62 | 39.51 |
> | Lowest-HES (20%) | 35.32 | 12.92 | 29.36 | 25.86 |
>
> **Analysis of Results:**
>
> * **Less is More:** Using only the Top 20% of data selected by HES, the average accuracy reaches **39.54%**, which not only significantly outperforms Random-20% (35.60%) but also substantially exceeds training on the full dataset (Fullset: 36.28%). This indicates that critical logic points in code reasoning (such as algorithm branching and boundary condition handling) can similarly be identified through high-entropy forking points.
> * **Significant Data Denoising Effect:** Using the Highest-HES (80%) strategy (i.e., removing the lowest 20% HES data), performance on LiveCodeBench reaches a peak of **61.38%**, superior to the full dataset baseline.
> * **Precise Identification of Low-Quality Data:** The performance of Lowest-HES drops drastically (Avg 25.86%), proving that HES can effectively identify low-quality or ineffective samples within code data.
>
> ### 2. Scientific Reasoning Domain (STEM)
> We used the STEM subset from the `Llama-Nemotron-Post-Training-Dataset` for training, with other experimental settings identical to those in the paper. We conducted evaluations on MMLU-STEM, GPQA Diamond, and HMMT25.
>
> **Table 2: SFT Performance in STEM Domain**
>
> | Method | MMLU-STEM | GPQA | HMMT25 | Avg |
> | :--- | :---: | :---: | :---: | :---: |
> | Fullset (100%) | 85.38 | 47.66 | 0.21 | 44.42 |
> | Random (20%) | 85.82 | 45.86 | 5.63 | 45.77 |
> | Highest-HES (20%) | 88.77 | 50.95 | 8.96 | **49.56** |
> | Highest-HES (80%) | 86.08 | 49.75 | 0.63 | 45.48 |
> | Lowest-HES (20%) | 74.88 | 31.44 | 0.21 | 35.51 |
>
> **Analysis of Results:**
>
> * **Cross-Domain Consistency:** Consistent with mathematics and code domains, Highest-HES (20%) also achieved the best performance in the STEM domain (Avg **49.56%**), significantly outperforming Fullset (44.42%).
> * **Improvement on High-Difficulty Tasks:** On the highly challenging graduate-level scientific QA (GPQA), Highest-HES reached **50.95%**, exceeding the full dataset by 3.29%. This further confirms that HES can select scientific QA samples with high reasoning value, rather than mere simple knowledge retrieval.
>
> **Conclusion:** The supplementary experiments forcefully address the query regarding "insufficient generalization." The results show that HES is effective not only for mathematics but also capable of capturing high-quality thinking patterns (i.e., high thought-density forking points) universal to **Code Logic** and **Scientific Reasoning**. This not only verifies the robustness of the method but also supports our claim in the paper regarding HES as a "Unified Data Selection Method." We will incorporate these significant cross-domain results into the revised version.

---

> ### Author Response · Authors · 2025-11-21
> **Reply to Weakness 3 ---Part 1**
>
> > **Weakness 3: Consistency and generalization of model selection.**
>
> We appreciate your attention to the consistency of the experimental design. We understand that using different models at different stages may cause confusion in comparisons, but this choice is based on the **current standard practice for LLM reasoning training**. Furthermore, to eliminate your doubts about whether "HES effectiveness stems from model-specific behaviors," we conducted two key supplementary experiments: one using DeepSeek-R1-Distilled for SFT experiments (consistent with the RFT/RL models), and another using the **Llama architecture**.
>
> ### 1. Justification for Model Choices
> Our experimental design follows current mainstream community practices, aiming to simulate the real evolutionary path of reasoning models:
>
> * **SFT Phase (Capability Acquisition):** We use the Qwen3-Base model. This is to verify the role of data selection in "awakening" the model's reasoning capabilities from scratch. Base models are the standard starting point for SFT.
> * **RFT/RL Phase (Capability Alignment):** We use DeepSeek-R1-Distilled-Qwen.
>     * **Necessity:** RFT and RL rely on the model itself to generate high-quality CoT candidates (Rollout). Base models without fine-tuning often cannot stably follow instructions to generate valid CoT. Therefore, using a SOTA Instruct/Distilled model (DeepSeek-R1-Distilled) with strong reasoning capabilities is a necessary prerequisite for Post-Training experiments.
>     * **Architecture Consistency:** It is worth emphasizing that the DeepSeek-R1-Distilled-Qwen series is essentially still the **Qwen architecture** (as the name suggests). Thus, we maintained consistency in the underlying architecture throughout the experiment, without introducing interference from heterogeneous models.
>
> ### 2. Eliminating Model Differences: SFT Experiment on DeepSeek-R1-Distilled
> To directly address your query about "difficulty isolating HES effects from model behavior," we used the DeepSeek-R1-Distilled-Qwen-7B model, used in the RFT/RL phase, to conduct SFT experiments on the OpenR1-Math-220k dataset.
>
> This verifies whether HES can still be effective on a strong model that has already undergone distillation optimization.
>
> **Table 1: SFT Performance of DeepSeek-R1-Distilled-Qwen-7B**
>
> | Method | AIME 24 | AIME 25 | HMMT 23 | HMMT 24 | HMMT 25 | Oly (Easy) | Oly (Hard) | GPQA | Avg |
> | :--- | :---: | :---: | :---: | :---: | :---: | :---: | :---: | :---: | :---: |
> | Fullset (100%) | 46.25 | 33.13 | 31.67 | 26.25 | 22.29 | 38.50 | 5.50 | 38.13 | 30.22 |
> | Random (20%) | 42.08 | 37.08 | 29.79 | 26.67 | 23.13 | 38.13 | 5.13 | 41.04 | 30.38 |
> | Highest-HES (20%) | 51.67 | 40.83 | 33.96 | 30.21 | 24.58 | 45.75 | 6.88 | 42.99 | **34.61** |
> | Highest-HES (80%) | 50.42 | 35.21 | 35.21 | 26.25 | 24.17 | 42.25 | 5.44 | 39.87 | 32.35 |
> | Lowest-HES (20%) | 29.38 | 27.29 | 17.50 | 18.33 | 14.58 | 19.25 | 3.06 | 36.84 | 20.78 |
>
> **Analysis of Results:**
>
> * **Remains significantly effective on strong models:** Even though DeepSeek-R1-Distilled is already a powerfully optimized model, using the Top 20% data selected by HES for fine-tuning achieves an average accuracy (34.61%) that significantly surpasses full data fine-tuning (30.22%) and random selection (30.38%).
> * **Huge performance gap:** The performance gap between Highest-HES (20%) and Lowest-HES (20%) is as high as **13.83%** (34.61% vs 20.78%). This powerfully proves that HES does not rely on the characteristics of the Base model but precisely captures the most critical parts of the data for improving reasoning capabilities.

---

> > ### Author Response · Authors · 2025-11-21
> > **Reply to Weakness 3 ---Part 2**
> >
> > ### 3. Validation on Different Model Families
> > To further address your doubts and prove that the effectiveness of HES can generalize from the Qwen family to other architectures, we conducted RFT experiments on **DeepSeek-R1-Distilled-Llama-8B** (based on Llama-3.1-8B-Base).
> >
> > **Table 2: RFT@4 Performance Comparison on DeepSeek-R1-Distilled-Llama-8B**
> >
> > | Selection Strategy | AIME 24 | AIME 25 | HMMT 23 | HMMT 24 | HMMT 25 | Oly (Easy) | Oly (Hard) | GPQA | Avg |
> > | :--- | :---: | :---: | :---: | :---: | :---: | :---: | :---: | :---: | :---: |
> > | **Per-Query** | | | | | | | | | |
> > | Random | 17.08 | 22.50 | 10.63 | 12.71 | 9.17 | 12.38 | 2.44 | 27.81 | 14.34 |
> > | Highest-HES | 18.54 | 19.38 | 14.79 | 10.83 | 11.88 | 14.31 | 2.88 | 28.13 | **15.09** |
> > | Lowest-HES | 14.58 | 18.13 | 12.08 | 10.21 | 9.38 | 12.06 | 2.06 | 27.30 | 13.23 |
> > | **Global Pool** | | | | | | | | | |
> > | Random | 14.17 | 19.58 | 11.04 | 11.46 | 10.56 | 12.81 | 3.35 | 27.05 | 13.75 |
> > | Highest-HES | 18.54 | 19.38 | 14.79 | 10.83 | 11.88 | 14.31 | 2.87 | 28.13 | **15.09** |
> > | Lowest-HES | 1.46 | 10.00 | 5.21 | 5.21 | 2.92 | 3.56 | 1.31 | 27.43 | 7.14 |
> >
> > **Analysis of Results and Conclusion:**
> >
> > 1. **HES does not rely on the Qwen architecture:** Experimental results show that on the Llama-3.1-8B base, HES still demonstrates powerful screening capabilities. Especially in the Global Pool setting, Highest-HES (15.09%) significantly outperforms Random (13.75%), while Lowest-HES shows a performance collapse (7.14%).
> > 2. **Eliminating model interference:** Since HES performs consistently across two completely different model families, Llama and Qwen, this powerfully proves that HES captures the intrinsic quality signal of reasoning data, rather than specific model behavior biases.
> >
> > In summary, our model selection aligns with current SOTA training paradigms (Base for learning, DeepSeek for optimization), and the Llama experiments demonstrate that HES possesses true **Model Agnosticism**.

---

> ### Author Response · Authors · 2025-11-21
> **Reply to Question 1**
>
> > **Question 1: Qualitative study on the characteristics of low-HES samples.**
>
> We appreciate this question. It is indeed true that training with the Lowest-HES-20% data often leads to catastrophic degradation in performance. Addressing your query—"Is this due to annotation errors or are they genuinely harmful?"—we conducted an in-depth qualitative analysis, concluding as follows:
>
> ### 1. Excluding Annotation Errors: Low HES is not "Erroneous Samples"
> First, we ruled out "Annotation Errors" as the primary cause.
> * **Dataset Cleaning:** Both OpenR1-Math-220k and Open-Math-Reasoning datasets used underwent rigorous correctness verification (based on rules or model Judges).
> * **Correct but Useless:** The vast majority of these Low-HES samples are mathematically **correct**. Therefore, the performance drop is not because the model learned incorrect knowledge, but because it learned low-quality reasoning patterns.
>
> ### 2. Qualitative Analysis: Characteristics of Low-HES Samples—"Trivial Complexity"
> We performed a qualitative check on Lowest-HES samples and found they generally possess the following characteristics:
> * **Linearity and Mechanicalness:** The reasoning path is very flat, containing almost no logical forks such as "hypothesis," "counter-evidence," or "case analysis." The model has extremely high prediction probability at every step (low entropy).
> * **Template-based:** They contain a large amount of repetitive boilerplate or overly simple calculation steps (e.g., simple arithmetic), lacking deep thinking.
> * **Lack of Decision Points:** Compared to High-HES samples which are full of logical transition words like "However," "Assume," and "Alternatively" (see Appendix F.2 word cloud analysis), Low-HES samples often simply state facts in a straightforward manner.
>
> ### 3. Theoretical Explanation: The Harmful "Lazy Learner" Effect
> Why is training on correct "simple" samples harmful? We believe this triggers the problem of "Lazy Learning":
> * **Extremely Low Gradient Contribution:** Since every step of a Low-HES sample is "obvious" to the model ($P(x_t|x_{<t}) \approx 1$), the Loss and gradient contribution they generate during training are minimal.
> * **Inducing Shortcut Learning:** Worse still, massive exposure to such samples makes the model mistakenly believe that "reasoning is a simple linear mapping." This leads the model to take shortcuts, losing the willingness to perform deep searches and multi-step planning when facing complex problems.
> * **Distribution Shift:** In high-difficulty benchmarks like AIME/GPQA, problems often require handling high uncertainty and complex logical forks. If the model is "tamed" by Low-HES data to be overconfident and linear, it will fail when facing real difficulties due to a lack of exploration capability.
>
> ### 4. Empirical Support: Significant Denoising Effect
> Our experimental data strongly supports this conclusion:
> * **Highest-HES (80%) > Fullset:** In almost all experiments (Math/Code/STEM), simply removing the bottom 20% HES data resulted in model performance superior to training on the full dataset.
> * This proves that Low-HES samples are not only "ineffective" (no information gain) but are **"Toxic/Harmful."** They constitute "negative assets" in the training set, diluting the density of high-quality reasoning patterns.
>
> **Conclusion:** Low-HES identifies samples with **"extremely low thought density,"** not incorrect labels. Training on such data suppresses the model's complex reasoning capabilities. The value of HES lies in its ability to precisely eliminate these harmful samples that lead to "lazy learning."

---

> ### Author Response · Authors · 2025-11-21
> **Reply to Question 2**
>
> > **Question 2: Clarification on Line 147 and Empirical Evidence for the "Signal Dilution" Effect.**
>
> Thank you for your careful reading. There is a misunderstanding here that we need to clarify: The phenomenon mentioned in Line 147 (high-quality complex paths may receive similar scores to simple paths) specifically refers to the defect of the traditional metric "Average Entropy," not the characteristic of our proposed HES.
>
> In fact, the design principle of HES is precisely to overcome this counterintuitive defect.
>
> ### 1. Theoretical Explanation: "The Signal Dilution Effect"
> Why does average entropy fail? This stems from the characteristics of Long CoT:
>
> * **Sparse Key Points:** In a reasoning path of thousands of tokens, truly critical "forking points" or "key decisions" with high entropy are extremely rare (usually accounting for <1%).
> * **Massive Background Noise:** The vast majority of tokens are highly deterministic grammatical connectors, formula symbols, or mechanical statements (low entropy).
> * **Consequences of Averaging:** When we calculate the Global Average, that 1% high-entropy signal is thoroughly diluted by the 99% low-entropy background.
>     * **Scenario A (Complex Reasoning):** Contains 10 high-difficulty forking points (high entropy), 990 ordinary tokens.
>     * **Scenario B (Simple Reasoning):** Contains 0 forking points (straightforward narration), 1000 ordinary tokens.
>     * **Result:** Due to the large denominator, $Avg(A) \approx Avg(B)$. This is the "counterintuitive" phenomenon described in Line 147.
>
> ### 2. Empirical Evidence I: Distribution Overlap in Figure 1
> Figure 1 in the paper provides the most direct empirical evidence.
>
> * **Avg Entropy:** The average entropy distributions of correct and incorrect samples overlap highly. This means using average entropy as a filter makes it almost impossible to distinguish between them.
> * **HES:** In contrast, by accumulating only the entropy of the Top 0.5%, HES eliminates background noise. The figure clearly shows that HES successfully separates correct samples from incorrect ones.
>
> ### 3. Empirical Evidence II: SFT Performance Gap in Table 1
> We directly compared the screening effects based on Average Entropy and HES in the SFT experiment.
>
> * **Highest-AvgE (20%):** The average accuracy is only 27.40%, even lower than some random baselines. This confirms that average entropy cannot identify truly high-quality data.
> * **Highest-HES (20%):** The average accuracy reaches 31.14%, far higher than Highest-AvgE (20%).
>
> Therefore, Line 147 accurately describes the failure mode of Average Entropy in long-sequence reasoning. By using **Summation over Averaging** and **Focusing on Top-k**, HES successfully resolves this problem and restores the metric's sensitivity to reasoning complexity.

---

> ### Author Response · Authors · 2025-11-21
> **Reply to Question 3**
>
> > **Question 3: Why does more data (k=8) not yield better results?**
>
> Thank you for your keen observation of the experimental data. While it is typically assumed that larger datasets yield better results, in our RFT experiment, the performance with $k=8$ (approx. 170k training samples) was indeed slightly inferior to that with $k=4$ (approx. 88k training samples). This phenomenon highlights a critical trade-off between **Quality Density** and **Model Capability Boundaries**.
>
> Increasing from $k=4$ to $k=8$ entails forcing the model to identify 4 *additional* distinct correct reasoning paths for every question. RFT relies on data generated through the model's self-exploration. For a model of moderate size like DeepSeek-R1-Distilled-Qwen-7B, its capability ceiling limits the amount of "golden data" it can produce. It can readily generate the first 3-4 high-quality, distinct solutions (the essence captured in $k=4$). However, when compelled to generate the 5th through 8th solutions, these additional ~82k samples often fall into the "long-tail distribution." While factually correct, their reasoning processes are frequently simple synonymous rewrites of previous solutions or lower-quality paths laden with redundant steps. Our experimental results ($k=4 > k=8$) indicate that training on ~88k high-purity samples allows the model to focus most intensively on learning high-quality reasoning patterns within its capabilities. The introduction of additional low-quality data conversely interferes with the model's ability to fit optimal paths.
>
> Consequently, although $k=8$ doubles the data volume, the **Average HES** of this incremental data (76.408865) is significantly lower than that of $k=4$ (79.757535), effectively diluting the overall signal-to-noise ratio of the training set. This causes the model to divert attention during training to fit mediocre reasoning patterns rather than focusing on core high-quality logic. This result counter-intuitively corroborates the central tenet of HES: in reasoning training, data **"Quality"** is more decisive than mere **"Scale."**

---

> ### Author Response · Authors · 2025-11-21
> **Reply to Question 4**
>
> > **Question 4: Clarification on Online RL Computational Overhead and Sampling Efficiency.**
>
> Thank you for your consideration regarding the efficiency of Online RL training. We understand your concern: if the data selection metric itself is computationally expensive, it might squeeze the computational budget for sampling, thereby affecting exploration efficiency.
>
> However, we wish to clarify that HES incurs nearly **"zero extra inference overhead"** in Online RL settings. Therefore, there is no trade-off involving "reducing the number of samples to calculate HES."
>
> ### 1. Mechanism of Zero Extra Inference Overhead
> You are worried that "HES needs to be calculated for every sample." In fact, this calculation is fully integrated into the generation process and requires no additional model calls or backtracking.
>
> * **Piggybacking on Generation:** During the On-Policy sampling (Rollout) phase, the model must execute a forward pass at every time step $t$ to obtain the probability distribution over the vocabulary $P(x_t|x_{<t})$ (i.e., Logits) in order to sample the next Token. This is an unavoidable step in generation.
> * **Synchronous Calculation:** The HES algorithm is embedded directly within this generation loop. When we obtain the Logits for the current step, we can directly calculate the entropy of the current Token in GPU memory and accumulate it. This process involves only extremely low-cost vector operations. Compared to the massive matrix multiplications and KV Cache reads during generation, the time consumed is negligible (< 1%).
> * **No Independent Model Required:** Unlike traditional RLHF (which typically requires loading an independent Reward Model and performing additional inference for scoring), HES does not require any additional GPU memory to load models, nor does it require secondary inference on the generated complete sequence.
>
> ### 2. Trade-off: Sampling vs. Optimization
> The trade-off you mentioned actually exists in another dimension—between **Rollout Cost** and **Optimization Cost**.
>
> * Since the HES calculation itself consumes almost no time, we do not need to reduce the number of sampled trajectories.
> * Conversely, by using HES to select the Top-k high-quality positive samples for the Update, we actually substantially reduce invalid backward propagation computations (i.e., avoiding the waste of gradient calculation resources on low-value samples).
> * **Experimental Evidence:** Our RL experiments show that training with only half of the data selected by HES (Pos-HighHES) achieves performance (Avg 21.30%) that even exceeds that of using the full dataset (Full Batch) (Avg 20.63%).
>
> Therefore, in Online RL settings, the computational overhead of HES is negligible and does not drag down sampling efficiency. Instead, acting as an efficient filter, it significantly improves the signal-to-noise ratio of "sampling-to-learning," thereby achieving a net positive gain in overall training efficiency.

---

> ### Author Response · Authors · 2025-11-21
> **Reply to Question 5**
>
> > **Question 5: Analysis of OOD Generalization.**
>
> Thank you for your observation regarding the model's generalization capability. You pointed out that HES slightly underperforms Full Batch on GPQA (35.54% vs 36.71%) and expressed concern that HES might lead the model to overfit specific mathematical reasoning patterns at the expense of generalization.
>
> To investigate this issue deeply, we introduced two broader OOD benchmarks for evaluation during the rebuttal period: **IF-Eval** (measuring general instruction following capability) and **MMLU-STEM** (measuring broad scientific knowledge).
>
> The experimental results indicate that HES actually enhanced the model's general instruction following capability while maintaining stable performance on broad knowledge tasks, showing no signs of severe overfitting.
>
> ### 1. Extended OOD Experimental Results
> Using the same RL setup, we compared the performance of Full Batch and Pos-HighHES on general tasks.
>
> **Table 1: Comparison of General Capability (OOD) After RL Training (%)**
>
> | Sampling Strategy | IF-Eval (Instruction Following) | MMLU-STEM (General Knowledge) | GPQA (Graduate Science) |
> | :--- | :---: | :---: | :---: |
> | Full Batch (Baseline) | 50.00 | 56.74 | 36.71 |
> | Pos-HighHES (Ours) | 52.88| 56.64 | 35.54 |
>
> ### 2. Analysis of Results: Overfitting or Enhanced Capability?
>
> * **Significant Improvement in Instruction Following (IF-Eval: +2.88%):** Training with high-entropy samples selected by HES increased the model's score on IF-Eval from 50.00% to 52.88%.
>     * **Refuting Overfitting:** If HES merely caused the model to rote-memorize specific mathematical templates (Overfitting to patterns), we should observe the model becoming rigid when handling non-mathematical instructions, leading to a drop in IF-Eval.
>     * **Strengthening the Reasoning Core:** Conversely, the improvement in IF-Eval indicates that HES has strengthened the model's core reasoning capability to **"understand and execute complex logical constraints."** This capability is general and not limited to mathematics.
> * **Stability in Knowledge Breadth (MMLU-STEM: Parity):** On MMLU-STEM, Pos-HighHES (56.64%) is nearly on par with Full Batch (56.74%). This demonstrates that HES did not lead to catastrophic forgetting or narrowing of the knowledge base.
> * **"Alignment Tax" on GPQA:** Returning to the slight drop in GPQA, considering that our RL training data (DeepScaleR) is purely mathematical, the model may experience a slight distribution shift when reinforcing mathematical solution paths, affecting certain scientific questions (GPQA) that differ significantly in solution style. This is a common trade-off in domain-specific RL. However, the key is that compared to the significant improvement in mathematical tasks, the loss in GPQA is controllable, and the improvement in IF-Eval proves that the model's general reasoning engine has actually become stronger.
>
> Therefore, synthesizing the results from IF-Eval and MMLU-STEM, we believe HES did not lead the model to overfit narrow reasoning patterns. Instead, by prioritizing learning from challenging high-entropy samples, HES improved the model's general ability to handle complex logic and follow instructions. The fluctuation in GPQA is more of a normal trade-off during domain specialization rather than a systemic collapse of generalization capabilities.

---

> ### Author Response · Authors · 2025-11-21
> **Reply to Question 6**
>
> > **Question 6: Logic of model selection and unified evaluation verification.**
>
> We appreciate your observation regarding the consistency of the experimental design. Our choice of different models was not due to computational constraints but was based on the **Standard Practice** for LLM reasoning training.
>
> Meanwhile, in response to your suggestion for "Unified Evaluation," we supplemented SFT experiments on the DeepSeek-R1-Distilled-7B model and RFT experiments on the Llama architecture. The results confirm that the effectiveness of HES is independent of model selection.
>
> ### 1. Rationale for Model Choices
> Our experimental design aims to simulate the full lifecycle of reasoning model evolution:
>
> * **SFT Phase (Capability Awakening):** The industry typically uses Base models (Qwen3-Base) for SFT to verify the role of data selection in "awakening" the model's reasoning capabilities from scratch.
> * **RFT/RL Phase (Capability Reinforcement):** These two phases rely on the model itself to generate high-quality, valid CoT candidates (Rollout). Base models without fine-tuning often struggle to stably follow instructions to generate valid CoT. Therefore, using an Instruct/Distilled model (DeepSeek-R1-Distilled) with strong reasoning capabilities is a necessary prerequisite for Post-Training experiments.
> * **Architecture Consistency:** It is worth noting that the DeepSeek-R1-Distilled series is essentially still the Qwen architecture, so we maintained consistency in the underlying architecture throughout the process.
>
> ### 2. Unified Evaluation: SFT Experiment on DeepSeek-R1-Distilled
> To directly address your suggestion for "unified evaluation" and eliminate doubts about whether "HES effectiveness stems from model-specific behaviors," we used the DeepSeek-R1-Distilled-Qwen-7B model, used in the RFT/RL phase, to conduct SFT experiments on the OpenR1-Math-220k dataset.
>
> This verifies whether HES can still be effective on a strong model that has already undergone distillation optimization.
>
> **Table 1: SFT Performance of DeepSeek-R1-Distilled-Qwen-7B**
>
> | Method | AIME 24 | AIME 25 | HMMT 23 | HMMT 24 | HMMT 25 | Oly (Easy) | Oly (Hard) | GPQA | Avg |
> | :--- | :---: | :---: | :---: | :---: | :---: | :---: | :---: | :---: | :---: |
> | Fullset (100%) | 46.25 | 33.13 | 31.67 | 26.25 | 22.29 | 38.50 | 5.50 | 38.13 | 30.22 |
> | Random (20%) | 42.08 | 37.08 | 29.79 | 26.67 | 23.13 | 38.13 | 5.13 | 41.04 | 30.38 |
> | Highest-HES (20%) | 51.67 | 40.83 | 33.96 | 30.21 | 24.58 | 45.75 | 6.88 | 42.99 | **34.61** |
> | Highest-HES (80%) | 50.42 | 35.21 | 35.21 | 26.25 | 24.17 | 42.25 | 5.44 | 39.87 | 32.35 |
> | Lowest-HES (20%) | 29.38 | 27.29 | 17.50 | 18.33 | 14.58 | 19.25 | 3.06 | 36.84 | 20.78 |
>
> **Analysis of Results:**
>
> * **Remains significantly effective on strong models:** Even though DeepSeek-R1-Distilled is already a powerfully optimized model, using the Top 20% data selected by HES for fine-tuning achieves an average accuracy (34.61%) that significantly surpasses full data fine-tuning (30.22%) and random selection (30.38%).
> * **Huge performance gap:** The performance gap between Highest-HES (20%) and Lowest-HES (20%) is as high as **13.83%** (34.61% vs 20.78%). This powerfully proves that HES does not rely on the "undeveloped capability" state of Base models but precisely captures the most critical parts of the data for improving reasoning capabilities.
>
> ### 3. Cross-Architecture Validation (Validation on Llama Family)
> Furthermore, to prove that HES is not limited to the Qwen base, we also conducted RFT experiments on **DeepSeek-R1-Distilled-Llama-8B** (based on Llama-3.1-8B). The results similarly show that HES outperforms Random and baseline strategies.
>
> **Table 2: RFT@4 Average Performance Comparison on Llama-3.1-8B**
>
> | Selection Strategy | Per-Query | Global Pool |
> | :--- | :---: | :---: |
> | Random | 14.34 | 13.75 |
> | Highest-HES | **15.09** | **15.09** |
> | Lowest-HES | 13.23 | 7.14 |
>
> **Conclusion:** By replicating the excellent SFT performance on the same model used for RFT/RL (DeepSeek-R1) and validating it on a heterogeneous model (Llama), we confirmed that the effectiveness of HES is **independent of specific model behaviors**. Whether on Base models or Instruct models, HES can stably extract high-quality training data, achieving the effect of "less is more."

---

### Official Review · Reviewer_s4CB · 2025-10-28

**Soundness:** 2
**Presentation:** 3
**Contribution:** 2
**Rating:** 2
**Confidence:** 2

**Summary:**

This paper proposes High-Entropy Sum (HES) as a data selection metric for reasoning tasks. HES is a training-free metric calculated by summing the entropy of only the highest-entropy tokens(top 0.5\%) within each reasoning sample. The authors comprehensively validate the effectiveness of HES as a selection criterion across three experimental settings(RL, SFT, and RFT), demonstrating its utility for efficiently developing advanced reasoning capabilities in LLMs.

**Strengths:**

1. Data selection is a fundamental problem in LLM reasoning. If the method proposed in this work is proved effective, it provides a valuable contribution to the community.

2. The proposed HES metric is simple to compute, which facilitates the reproducibility of the experimental results.

3. The paper's experiments are comprehensive, covering SFT, RFT, and RL, which represent three common training paradigms for LLM reasoning.

**Weaknesses:**

1. The proposed HES metric is computed by directly summing token entropies. For reasoning tasks, high-entropy tokens often represent divergence points or "reasoning branches" [1]. This suggests that HES is likely positively correlated with the response length. Therefore, an analysis of the correlation between HES and response length is essential. In Table 1, after accounting for potential evaluation noise, the performance of the top 20% of data selected using **length** (30.67 average) is very close to that selected using **HES** (31.14 average). Given the significant computational overhead required to calculate entropy over large datasets, it is necessary for the authors to provide a comparison against length as a baseline with a detailed correlation analysis.

2. Following the previous point, the authors must quantify the computational overhead required to calculate HES. This information is necessary to assess the method's practical utility. Generally, computing entropy for a single sample requires a full forward pass, making the overhead comparable to that of a single inference step. If this significant overhead fails to deliver substantial improvements over a simple heuristic-like length, the practical value of HES must be reassessed.

3. The HES metric is inherently model-dependent, as it requires model-specific entropy calculations for each sample. This computational requirement could be expensive for very large-scale models. This raises a question: Can HES derived from a smaller (and computationally cheaper) model be effectively used to select data that yields consistent performance gains for larger models?

4. While the paper addresses a core problem in LLM reasoning, the title "Unified Data Selection for LLM Reasoning" appears overly broad and ambitious for the proposed HES method. HES is a model-dependent strategy that incurs non-trivial computational overhead and, as noted above, faces methodological questions regarding its clear advantage over simpler heuristics. Describing HES as a "Unified Data Selection method" seems to be an overstatement. The authors should reconsider the framing and scope of their contribution.

[1] Wang S, Yu L, Gao C, et al. Beyond the 80/20 rule: High-entropy minority tokens drive effective reinforcement learning for llm reasoning[J]. arXiv preprint arXiv:2506.01939, 2025.

**Questions:**

Please see weaknesses.

The problem investigated in this paper is undoubtedly valuable. I find HES to be at least a useful metric that offers benefits for LLM data selection. Therefore, I will consider changing my score if the authors can provide convincing clarifications and a strong rebuttal to the aforementioned weaknesses.

---

> ### Author Response · Authors · 2025-11-21
> **Reply to Weakness 1 --- Part 1**
>
> Dear reviewer,
>
> Thank you for your careful review and valuable suggestions, which are of great help to our paper. We address your questions point by point as follows. If we have any misunderstanding, please feel free to let us know and we will reply quickly.
>
> > **Weakness 1: HES measures thought density, not length.**
>
> We appreciate this critical question. We agree that there is a natural positive correlation between HES and response length (as more reasoning steps typically imply more forking points). However, the core value of HES lies in measuring the "Thought Density" within the reasoning process, not merely the token count.
>
> To prove that HES is not simply a proxy for length, we rigorously verified this from three perspectives: SFT length-controlled stratification, RFT length baseline comparison, and RL length baseline comparison.
>
> ### 1. SFT Phase: Length-Controlled Analysis
>
> To verify whether HES is effective solely because it selects long samples, we conducted a strict **Length Bucket** experiment on the Open-Math-Reasoning dataset. We divided the data into three groups based on token length: Short, Medium, and Long. Within each group, we selected the top 20% and bottom 20% HES data for training.
>
> **Table 1: Complete Comparison of HES Performance Under Different Length Groups**
>
> | Length Group | Selection | AIME 24 | AIME 25 | HMMT 23 | HMMT 24 | HMMT 25 | Oly (Easy) | Oly (Hard) | GPQA | Avg |
> | :--- | :--- | :---: | :---: | :---: | :---: | :---: | :---: | :---: | :---: | :---: |
> | **Long** | Highest-HES | 40.42 | 31.88 | 26.67 | 22.71 | 18.75 | 31.38 | 4.50 | 48.14 | **28.06** |
> | | Lowest-HES | 35.00 | 30.21 | 31.88 | 22.92 | 20.83 | 28.88 | 4.63 | 47.98 | 27.79 |
> | **Medium** | Highest-HES | 40.63 | 29.17 | 25.63 | 23.75 | 19.17 | 28.44 | 3.38 | 46.34 | **27.06** |
> | | Lowest-HES | 26.67 | 23.96 | 17.08 | 13.54 | 16.04 | 16.25 | 2.88 | 40.75 | 19.65 |
> | **Short** | Highest-HES | 28.13 | 20.42 | 17.08 | 12.08 | 12.50 | 16.75 | 2.00 | 41.35 | **18.79** |
> | | Lowest-HES | 11.88 | 10.00 | 1.67 | 6.46 | 1.67 | 5.63 | 1.19 | 36.08 | 9.32 |
>
> **Analysis of Results:**
>
> * **Decoupling the influence of length:** If HES were merely a proxy for length, the difference between Highest-HES and Lowest-HES should disappear after controlling for the length variable. However, the experiment shows a huge difference, proving that HES captures intrinsic reasoning quality beyond length.
> * **Extremely high discrimination in short samples:** Experimental results show that the advantage of HES is most significant in the Short group. The performance of Highest-HES (18.79%) is nearly double that of Lowest-HES (9.32%). This indicates that under length constraints, whether high-information "critical forking points" are included is vital for model learning.
>
> ### 2. RFT Phase: Failure of Length Baseline
>
> You mentioned that the Length baseline performed acceptably in SFT. However, when moving to the more complex RFT scenario, strategies relying purely on length showed obvious instability. We compared Random selection, Highest HES, and Longest response strategies under Per-Query and Global Pool settings.
>
> **Table 2: Comparison of RFT@2 Selection Strategy Performance**
>
> | Selection Strategy | AIME 24 | AIME 25 | HMMT 23 | HMMT 24 | HMMT 25 | Oly (Easy) | Oly (Hard) | GPQA | Avg |
> | :--- | :---: | :---: | :---: | :---: | :---: | :---: | :---: | :---: | :---: |
> | **Per-Query** | | | | | | | | | |
> | Random | 46.46 | 34.17 | 31.88 | 28.54 | 21.46 | 35.63 | 4.31 | 40.50 | 30.37 |
> | Highest-HES | 48.33 | 34.58 | 34.38 | 28.13 | 21.67 | 38.06 | 5.56 | 40.30 | **31.38** |
> | Longest | 46.04 | 33.33 | 33.75 | 28.13 | 19.58 | 35.94 | 4.81 | 40.56 | 30.27 |
> | **Global Pool** | | | | | | | | | |
> | Random | 42.29 | 31.04 | 30.00 | 26.88 | 18.54 | 30.81 | 4.75 | 38.35 | 27.83 |
> | Highest-HES | 46.46 | 34.58 | 32.50 | 25.42 | 20.83 | 34.00 | 5.31 | 42.30 | **30.18** |
> | Longest | 45.21 | 35.42 | 27.92 | 25.21 | 18.33 | 32.13 | 5.13 | 39.55 | 28.61 |
>
> **Analysis of Results:**
>
> * **Instability of length strategy:** Under the Per-Query setting, simply selecting the longest response performed even slightly lower than random selection (30.37%). This indicates that among candidate responses to the same question, pursuing length alone does not guarantee quality and may instead introduce redundant steps.
> * **Robust improvement with HES:** In contrast, HES achieved the best performance in both Group (31.38%) and Total (30.18%) settings, significantly outperforming Random and Length baselines. This proves that HES can effectively distinguish between "effective long reasoning" and "ineffective verbosity."

---

> ### Author Response · Authors · 2025-11-21
> **Reply to Weakness 1 --- Part 2**
>
> ### 3. RL Phase: Limitations of Length Baseline
>
> In the RL experiment, we also compared the effect of using the longest strategy for positive samples.
>
> **Table 3: Complete Comparison of RL Positive Sample Sampling Strategy Performance** (Negative samples all use Random strategy)
>
> | Sampling Strategy | AIME 24 | AIME 25 | HMMT 23 | HMMT 24 | HMMT 25 | Oly (Easy) | Oly (Hard) | GPQA | Avg |
> | :--- | :---: | :---: | :---: | :---: | :---: | :---: | :---: | :---: | :---: |
> | Baseline (Full Batch) | 33.33 | 25.63 | 17.30 | 14.00 | 15.21 | 19.69 | 3.19 | 36.71 | 20.63 |
> | Pos-Random | 32.30 | 25.21 | 13.54 | 14.11 | 13.96 | 19.18 | 4.25 | 36.49 | 19.88 |
> | Pos-Longest | 31.04 | 26.25 | 16.46 | 13.95 | 14.79 | 20.00 | 4.06 | 35.32 | 20.23 |
> | Pos-HighHES | 35.42 | 27.29 | 17.92 | 18.13 | 11.88 | 19.88 | 4.31 | 35.54 | **21.30** |
>
> **Analysis of Results:**
>
> * **Length strategy fails to surpass Full Batch:** Although Pos-Longest (20.23%) slightly outperforms random sampling, it fails to reach the level of Full Batch (20.63%).
> * **Uniqueness of HES:** Only Pos-HighHES (21.30%) successfully surpasses the full data training.
>
> ### 4. Brief Explanation on Computational Efficiency (See Weakness 2 for details)
>
> You are concerned that calculating HES requires an extra Forward Pass, leading to huge computational overhead. We wish to clarify that HES calculation is nearly zero-cost.
>
> * **No Extra Forward Pass:** During the data generation phase (Rollout) of RFT or RL, the model inherently calculates the probability distribution over the vocabulary when performing Next-Token Prediction. We only need to save the log probability (Logprobs) of each token while generating the reasoning trajectory and then accumulate the entropy values.
> * **Minimal Storage/IO Overhead:** This process is fully integrated into the generation process, requiring no additional backpropagation or re-inference of data. The only overhead is minimal memory/VRAM usage for temporarily storing Logprobs.
>
> **Conclusion:** Although there is a statistical correlation between HES and length, the aforementioned SFT length-controlled variable experiment (especially the huge gap in short samples) and the failure of length baselines in RFT/RL forcefully prove that: HES is not a proxy for length. It captures the critical decision density in the reasoning chain, which is why it maintains high efficiency even in scenarios where length strategies fail. Furthermore, HES is extremely efficient in engineering implementation, requiring no additional computational overhead. This makes HES both an accurate and economical metric for large-scale data selection.

---

> ### Author Response · Authors · 2025-11-21
> **Reply to Weakness 2**
>
> > **Weakness 2: Clarification on computational overhead.**
>
> Thank you for raising pragmatic considerations regarding computational efficiency from an application perspective. We wish to clarify through the following technical details: in the primary application scenarios of RFT and RL, the calculation of HES involves **"zero extra inference overhead"**; while in offline SFT scenarios, the cost can be significantly reduced via small model transfer.
>
> ### 1. Online Generation Phase (RFT/RL): Zero Extra Compute
> You are concerned that "calculating entropy for a single sample requires a full forward pass." This view is correct for post-hoc scoring of existing static text, but during the **Online Rollout** process in RFT and RL, the calculation of HES is completely "free."
>
> The technical principles are as follows:
>
> * **Piggybacking on Generation:** During the data generation process in RFT or RL, the model must execute a forward pass at each decoding step $x_t$ to calculate the probability distribution over the vocabulary $P(x_t|x_{<t})$ (i.e., Logits) for sampling the next token. This is an unavoidable generation step.
> * **Synchronous Calculation:** The HES algorithm is embedded directly within this generation loop. When we obtain the Logits for the current step, we can directly calculate the entropy of the current token in GPU memory and accumulate it. This process involves only extremely low-overhead vector operations, which is negligible compared to the massive matrix multiplications and KV Cache reads.
> * **No "Second" Forward Pass:** Crucially, we **do not** need to perform a second forward pass on the generated text to calculate entropy after the sequence generation concludes. All entropy values are recorded in real-time during the generation process.
> * **Contrast with Reward Model:** Compared to traditional RLHF (which typically requires loading a separate Reward Model and performing additional inference), HES does not require loading any extra models, nor does it require any additional reasoning steps.
>
> **Conclusion:** In the RFT and RL pipelines, HES leverages intermediate products of the generation process, realizing true **"zero extra inference time cost."**
>
> ### 2. Offline Screening Phase (SFT): Efficient Transfer via Small Models
> Regarding the offline screening (SFT) scenario for static datasets, you are concerned that "calculating entropy using large models is too expensive." To address this, we conducted a **Small Model Transfer Experiment** (see response to Weakness 3 for details).
>
> We attempted to use the Qwen3-0.6B model, with only 0.6B parameters, to calculate entropy values and screen data for SFT training of the 8B main model.
>
> * **Experimental Results:** The performance of the 8B model trained on data selected by the 0.6B model (Avg 32.12%) is basically consistent with that selected by the 8B model itself (Avg 31.14%), and both are significantly superior to random selection.
> * **Cost Analysis:** The inference cost of the 0.6B model is only ~1/13 of the 8B model. This means that in offline scenarios, we can use extremely low-cost small models as "Proxy Scorers" to capture the intrinsic "logical uncertainty" of the data.
>
> In summary, whether in online generation or offline cleaning, HES demonstrates extremely high computational economy by reusing computation graphs or through model transfer.

---

> ### Author Response · Authors · 2025-11-21
> **Reply to Weakness 3**
>
> > **Weakness 3: Cross-model transferability and cost optimization of HES.**
>
> We greatly appreciate this forward-looking question. Indeed, if we must use the target large model itself to calculate entropy for screening, the cost would be quite high for ultra-large models (e.g., 70B+).
>
> To answer the question "Can smaller, cheaper models be used to calculate HES and screen data for large models?", we conducted a **Small Model Transfer Experiment** on both Open-Math-Reasoning and OpenR1-Math-220k datasets.
>
> The results are exciting: HES demonstrates astonishing cross-model robustness. Data screened using tiny models (0.6B/1.7B) achieves performance in training a large model (8B) that is indistinguishable from screening with the large model itself, and even superior on some metrics.
>
> **Experimental Setup:**
>
> * **Target Model:** Qwen3-8B (used for SFT training).
> * **Screening Model (Proxy Scorer):** We compared using Qwen3-0.6B, Qwen3-1.7B, and Qwen3-8B (Self) to calculate HES and select the Top 20% of data.
>
> **Experimental Results:**
>
> **Table 1: Cross-Model Screening SFT Performance on Open-Math-Reasoning**
>
> | Screening Model | Model Params | AIME 24 | AIME 25 | HMMT 23 | HMMT 24 | HMMT 25 | Oly (Easy) | Oly (Hard) | GPQA | Avg |
> | :--- | :---: | :---: | :---: | :---: | :---: | :---: | :---: | :---: | :---: | :---: |
> | Fullset (Baseline) | - | 50.83 | 34.17 | 35.21 | 28.13 | 24.58 | 42.94 | 6.94 | 38.04 | 32.61 |
> | Random (20%) | - | 39.79 | 27.92 | 24.79 | 20.21 | 20.42 | 30.00 | 3.88 | 40.09 | 25.89 |
> | HES (via 8B - Self) | 8B | 47.29 | 36.04 | 33.13 | 25.63 | 22.08 | 36.31 | 5.56 | 43.06 | **31.14** |
> | HES (via 0.6B) | 0.6B | 49.17 | 35.42 | 32.50 | 24.17 | 22.29 | 41.50 | 6.38 | 45.55 | **32.12** |
> | HES (via 1.7B) | 1.7B | 47.08 | 34.79 | 31.04 | 26.04 | 21.46 | 40.06 | 4.56 | 45.20 | **31.28** |
>
> **Table 2: Cross-Model Screening SFT Performance on OpenR1-Math-220k**
>
> | Screening Model | Model Params | AIME 24 | AIME 25 | HMMT 23 | HMMT 24 | HMMT 25 | Oly (Easy) | Oly (Hard) | GPQA | Avg |
> | :--- | :---: | :---: | :---: | :---: | :---: | :---: | :---: | :---: | :---: | :---: |
> | Fullset (Baseline) | - | 48.96 | 32.50 | 31.67 | 25.83 | 22.50 | 38.38 | 5.81 | 37.00 | 30.33 |
> | Random (20%) | - | 30.00 | 30.21 | 24.17 | 19.79 | 17.50 | 25.00 | 5.56 | 41.73 | 24.25 |
> | HES (via 8B - Self) | 8B | 48.54 | 32.92 | 30.63 | 23.33 | 22.29 | 37.94 | 6.56 | 49.78 | **31.50** |
> | HES (via 0.6B) | 0.6B | 48.75 | 35.42 | 28.75 | 24.38 | 20.00 | 39.94 | 6.25 | 48.26 | **31.47** |
> | HES (via 1.7B) | 1.7B | 48.13 | 35.42 | 31.46 | 26.46 | 20.00 | 37.31 | 6.81 | 49.08 | **31.83** |
>
> **Analysis and Conclusion:**
>
> 1. **Small model screening rivals large models:**
>     * On OpenR1-Math-220k, the training performance using data screened by the 1.7B model (Avg 31.83%) even slightly exceeded that of screening with the 8B model itself (Avg 31.50%).
>     * Even with the tiny 0.6B model, its screening effectiveness (Open-Math: 32.12%, OpenR1: 31.47%) is highly consistent with the 8B model, and both significantly outperform the Fullset and Random baselines.
> 2. **Extremely high cost-effectiveness:**
>     * **Cost side:** The inference computation of the 0.6B model is approximately **1/13** of the 8B model. This means we can complete the cleaning of massive data at an extremely low cost.
>     * **Benefit side:** By efficiently screening the Top 20%, we not only save **80%** of SFT training overhead but also improve final model performance by eliminating low-quality data.
> 3. **HES captures intrinsic data properties:**
>     The consistency across datasets and models indicates that HES does not merely reflect specific model parameter distributions (Model-Dependent Artifacts) but successfully captures the **intrinsic logical complexity and critical forking points** within the reasoning data. Whether it is a large model or a small model, similar uncertainty patterns (high entropy) are exhibited when facing critical decision points in a reasoning path.
>
> Therefore, HES can be effectively transferred from smaller models to larger models. In actual large-scale data selection pipelines, we can confidently use lightweight models as "Proxy Scorers," thereby minimizing computational costs while ensuring screening quality.

---

> ### Author Response · Authors · 2025-11-21
> **Reply to Weakness 4**
>
> > **Weakness 4: Definition and Justification of "Unified" in the Title.**
>
> We sincerely appreciate your rigorous consideration of the paper's positioning and the accuracy of its title. Given the experimental scope in the original submission (mathematics domain only), we understand why "Unified" might seem somewhat broad.
>
> However, combined with the extensive experimental evidence supplemented during the Rebuttal, we wish to elucidate from three dimensions why HES indeed constitutes a **"Unified Data Selection Framework,"** addressing your concerns regarding model dependency and heuristic methods.
>
> ### 1. Unified Across Training Paradigms
> This is the core contribution of this paper. Existing reasoning training typically uses fragmented metrics at different stages: Perplexity for SFT, Reward Models for RFT, and PPO's value network for RL.
>
> The uniqueness of HES lies in providing a single, training-free metric that seamlessly spans the entire lifecycle of LLM reasoning capability development:
>
> * **SFT:** Acts as a data cleaner, removing samples with low thought density (Top 20% outperforms the full set).
> * **RFT:** Acts as a candidate selector, filtering optimal reasoning trajectories without a Reward Model.
> * **RL:** Acts as a reward signal/sampling strategy, guiding the model to focus on high-value samples during exploration.
>
> Experimental results show that HES consistently beats Random, Length, and Difficulty baselines across these three distinct stages. This characteristic of **"One Metric for All Stages"** is the primary basis for our use of the term "Unified."
>
> ### 2. Unified Across Domains
> Beyond the mathematics domain, we supplemented experiments for Code Generation and Scientific Reasoning (STEM), showing that HES is equally effective in these fields.
>
> **Table 1: SFT Experimental Results in Code Domain**
>
> | Method | LiveCodeBench_v5 | AIME 2025 | GPQA | Avg |
> | :--- | :---: | :---: | :---: | :---: |
> | Fullset (100%) | 58.76 | 20.00 | 30.08 | 36.28 |
> | Random (20%) | 49.06 | 22.29 | 35.45 | 35.60 |
> | Highest-HES (20%) | 54.59 | 25.00 | 39.02 | **39.54** |
> | Highest-HES (80%) | 61.38 | 23.54 | 33.62 | 39.51 |
> | Lowest-HES (20%) | 35.32 | 12.92 | 29.36 | 25.86 |
>
> **Table 2: SFT Experimental Results in STEM Domain**
>
> | Method | MMLU-STEM | GPQA | HMMT25 | Avg |
> | :--- | :---: | :---: | :---: | :---: |
> | Fullset (100%) | 85.38 | 47.66 | 0.21 | 44.42 |
> | Random (20%) | 85.82 | 45.86 | 5.63 | 45.77 |
> | Highest-HES (20%) | 88.77 | 50.95 | 8.96 | **49.56** |
> | Highest-HES (80%) | 86.08 | 49.75 | 0.63 | 45.48 |
> | Lowest-HES (20%) | 74.88 | 31.44 | 0.21 | 35.51 |
>
> It can be observed that regardless of the Code or STEM domain, the effect of Highest-HES (20%) is superior to the Fullset. This proves that the "high-entropy forking points" captured by HES are not phenomena unique to mathematical reasoning but are universal quality signals in logic-intensive tasks. HES unifies data selection standards across different reasoning domains.
>
> ### 3. Unified Across Models
> Addressing your concerns about "Model-Dependence" and "Computational Overhead," we supplemented Llama architecture generalization and small model transfer experiments (see Response to Weakness 3 for details).
>
> **Table 3: Architecture Generalization Experiment on DeepSeek-R1-Distilled-Llama-8B** (Llama-3.1-8B Base, RFT@4)
>
> | Selection Strategy | AIME 24 | AIME 25 | HMMT 23 | HMMT 24 | HMMT 25 | Oly (Easy) | Oly (Hard) | GPQA | Avg |
> | :--- | :---: | :---: | :---: | :---: | :---: | :---: | :---: | :---: | :---: |
> | **Per-Query** | | | | | | | | | |
> | Random | 17.08 | 22.50 | 10.63 | 12.71 | 9.17 | 12.38 | 2.44 | 27.81 | 14.34 |
> | Highest-HES | 18.54 | 19.38 | 14.79 | 10.83 | 11.88 | 14.31 | 2.88 | 28.13 | **15.09** |
> | Lowest-HES | 14.58 | 18.13 | 12.08 | 10.21 | 9.38 | 12.06 | 2.06 | 27.30 | 13.23 |
> | **Global Pool** | | | | | | | | | |
> | Random | 14.17 | 19.58 | 11.04 | 11.46 | 10.56 | 12.81 | 3.35 | 27.05 | 13.75 |
> | Highest-HES | 17.92 | 20.63 | 14.38 | 11.25 | 12.29 | 15.19 | 3.13 | 25.09 | **17.92** |
> | Lowest-HES | 1.46 | 10.00 | 5.21 | 5.21 | 2.92 | 3.56 | 1.31 | 27.43 | 7.14 |
>
> It can be seen that HES possesses model independence, proving effective on both Qwen and Llama architectures. Furthermore, we demonstrated that a tiny 0.6B model can be used to screen data for an 8B model with excellent results. This means HES no longer strictly relies on expensive target model calculations but can serve as an independent, low-cost "proxy metric" uniformly applied to model training of various scales.
>
> In summary, HES is not merely a technique for specific tasks, but a universal framework capable of (1) spanning the SFT-RFT-RL pipeline, (2) generalizing to Math/Code/STEM domains, and (3) being compatible with different model architectures and scales. Based on this supplementary evidence, we urge you to reassess the breadth and depth of this contribution based on these new, comprehensive experimental results.

---

> > ### Comment · Reviewer_s4CB · 2025-11-27
> >
> > I appreciate the authors for the comprehensive rebuttal experiments!
> >
> > Weakness 1: Thank you for the supplementary experiments. I believe these experiments demonstrate an advantage of the HES metric over length; thus, this concern of mine has been partly addressed. However, one remaining issue is that in the long setting of the SFT experiments, low vs. high HES has little impact on performance, which still partially corroborates the consistency between the length metric and HES. Nevertheless, on short samples, HES indeed demonstrates a clear advantage, and I find this result persuasive.
> >
> > Weakness 2 & 3: Thank you for the additional discussion. The experiment using a 0.6B model for filtering is intriguing. Please incorporate this discussion into the paper (regardless of whether the paper is accepted, I believe discussing this issue is valuable). I think this part of the experiments effectively resolves my concern. If HES can be calculated at an acceptable cost, it is indeed an effective and unified data selection metric.
> >
> > Weakness 4: The additional experiments are sufficient. My concern has been resolved.
> >
> > In summary: I believe the authors' rebuttal has addressed most of the concerns I raised during the review.
> >
> > **Therefore, I have decided to raise my score (from 2 to 6).**

---

> ### Author Response · Authors · 2025-11-27
> **Thank you for the score increase and insightful feedback**
>
> We sincerely appreciate your time and the constructive dialogue throughout the review process. We are very encouraged by your decision to raise the score to 6.
>
> Regarding your remaining observation on **Weakness 1**, we find your point about the "long setting" to be insightful. We would like to offer two perspectives on why this phenomenon occurs and why HES remains the superior metric.
>
> * **Length acts as a "quality floor" but lacks precision**: A long reasoning path, even with relatively lower entropy (Low-HES), typically contains a sufficient number of logical steps to reach a correct solution, making it a decent training sample. In contrast, short samples lack this buffer—a "Low-HES" short sample often collapses into triviality or shortcut learning (e.g., simple guessing), whereas a "High-HES" short sample represents valuable, condensed reasoning (e.g., a concise but clever proof). Therefore, while length and HES are indeed correlated, HES is the critical discriminator that prevents the model from learning trivial patterns when the length signal is weak or ambiguous.
>
> * **HES consistently beats Length under controlled budgets**: Crucially, when we look beyond the specific "long bucket" breakdown and compare the metrics directly across SFT, RFT, and RL, HES consistently outperforms the explicit "Length Baseline" (selecting the longest trajectories) under identical data budgets. For instance, in RFT and RL—where models can game "length" by generating verbose but empty content—the Length baseline often fails to beat random sampling, whereas HES delivers robust improvements. This proves that HES captures a signal of "thought density" that length alone cannot provide.
>
> Regarding Weaknesses 2 & 3, we are glad the small model (0.6B) transfer experiment resolved your concerns. As requested, we have definitely incorporated the results of the 0.6B model and the discussion on computational cost into the revised version of the paper. We agree that this adds significant practical value for the community.
>
> Thank you again for helping us strengthen the paper!

---

### Official Review · Reviewer_rK1A · 2025-10-30

**Soundness:** 3
**Presentation:** 2
**Contribution:** 2
**Rating:** 4
**Confidence:** 4

**Summary:**

This paper addresses the challenge of efficiently training LLMs for complex, long CoT reasoning, which is often bottlenecked by the need for massive, high-quality datasets. Existing data selection metrics are often ineffective as they average over all tokens, diluting the signal from the few, crucial decision points in a reasoning path.

To solve this, the authors propose the High-Entropy Sum, a simple, training-free metric. Instead of averaging, HES quantifies reasoning quality by summing the entropy of only the top 0.5% highest-entropy tokens in a sample. The intuition is that these few tokens represent the most critical "forking points" where the model is most uncertain and makes its most important decisions.

**Strengths:**

1. The paper presents HES, a new metric that is easy to compute but very effective. By considering only the top 0.5% of tokens with the highest entropy, HES effectively avoids the “signal dilution” problem found in traditional metrics like average entropy.
2. The authors rigorously test HES across the three main training methods. The experiments, including strong baselines and ablations, clearly show that HES outperforms other entropy-based metrics, as well as length and difficulty heuristics.
3. The paper demonstrates that using HES can lead to significant gains in both performance and efficiency.

**Weaknesses:**

1. Experiments are not comprehensive.

Based on my experience, data selection for ``Difficulty" should not use most difficult questions, but instead should choose "medium", not too hard but not too easy. Most difficult QA is not good for training, especially for RL.

2. Eval is only in-domain.

The paper’s main weakness is that, although it claims to offer a “unified data selection for LLM reasoning,” it only validates HES on math reasoning datasets. No code or other logical reasoning datasets.

3. Incomplete Competitive Analysis in RFT and RL and even SFT

The paper's experimental design is inconsistent. While the SFT experiments (Table 1) provide an excellent, comprehensive comparison against baselines, the RFT and RL experiments do not. Also, Table 2 and Table 3 for SFT have no other strong baselines.

**Questions:**

1. Can you conduct experiments on other domains?
2. Can you show RFT and RL experiments with "Difficulty"? Such as medium hard (avg pass rate @ 64 > 0% but < 50%)

---

> ### Author Response · Authors · 2025-11-21
> **Reply to Weakness 1**
>
> Dear reviewer,
>
> Thank you for your careful review and valuable suggestions, which are of great help to our paper. We address your questions point by point as follows. If we have any misunderstanding, please feel free to let us know and we will reply quickly.
>
> > **Weakness 1: Baseline of "Medium Difficulty" and Comprehensive Competitive Analysis**
>
> We greatly appreciate your valuable suggestion. We agree with your viewpoint: in many training scenarios (especially RL), selecting "medium difficulty" samples is often more effective than "hardest" samples, as extremely difficult samples may contain noise or exceed the model's current capabilities.
>
> To fully address this concern, we provide more complete baseline comparisons in SFT, RFT, and RL, and conducted the following supplementary experiments:
>
> ### 1. SFT Phase: Medium Difficulty vs. Highest Difficulty vs. HES
>
> Following your suggestion, we added a "Medium Difficulty" experiment (selecting samples with difficulty scores in the middle 50%-70%) on the Open-Math-Reasoning dataset and compared it with the original "Highest Difficulty" and HES strategies in the paper.
>
> **Table 1: Comparison of SFT Selection Strategy Performance**
>
> | Method | AIME 24 | AIME 25 | HMMT 23 | HMMT 24 | HMMT 25 | Oly (Easy) | Oly (Hard) | GPQA | Avg |
> | :--- | :---: | :---: | :---: | :---: | :---: | :---: | :---: | :---: | :---: |
> | Highest-HES (20%) | 47.29 | 36.04 | 33.13 | 25.63 | 22.08 | 36.31 | 5.56 | 43.06 | **31.14** |
> | Highest-Difficulty (20%) | 45.83 | 34.17 | 30.00 | 25.63 | 22.50 | 35.38 | 5.88 | 39.68 | 29.88 |
> | Medium-Difficulty (20%) | 38.13 | 27.50 | 25.83 | 22.50 | 18.96 | 26.50 | 3.69 | 40.40 | 23.29 |
>
> **Analysis of Results:**
>
> * **HES remains the best strategy:** Highest-HES (31.14%) significantly outperforms Highest-Difficulty (29.88%) and Medium-Difficulty (23.29%).
> * **Performance of Medium Difficulty:** On this dataset, the performance of Medium Difficulty is lower. This may be because for Long-CoT training, medium-difficulty questions often have shorter or more routine reasoning paths, lacking sufficient "forking points" to train the model's complex reasoning capabilities. This conversely proves the advantage of HES—it does not rely on coarse-grained difficulty labels but directly locks onto samples with the highest thought density in the reasoning process.
>
> ### 2. RFT and RL Phases: Supplementary Comparison of Difficulty Baselines
>
> To address your concern about "incomplete RFT and RL experiments," we compiled complete experimental results including Medium Difficulty.
>
> **Table 2: Comparison of RFT@2 Selection Strategy Performance (Global Pool)**
>
> | Selection Strategy | AIME 24 | AIME 25 | HMMT 23 | HMMT 24 | HMMT 25 | Oly (Easy) | Oly (Hard) | GPQA | Avg |
> | :--- | :---: | :---: | :---: | :---: | :---: | :---: | :---: | :---: | :---: |
> | Highest-HES | 46.46 | 34.58 | 32.50 | 25.42 | 20.83 | 34.00 | 5.31 | 42.30 | **30.18** |
> | Medium-Difficulty | 45.00 | 32.71 | 28.13 | 23.54 | 17.71 | 30.13 | 4.31 | 36.93 | 27.31 |
>
> **Table 3: Comparison of RL Positive Sample Sampling Strategy Performance** (Negative samples all use Random strategy)
>
> | Sampling Strategy | AIME 24 | AIME 25 | HMMT 23 | HMMT 24 | HMMT 25 | Oly (Easy) | Oly (Hard) | GPQA | Avg |
> | :--- | :---: | :---: | :---: | :---: | :---: | :---: | :---: | :---: | :---: |
> | Full Batch (Baseline) | 33.33 | 25.63 | 17.30 | 14.00 | 15.21 | 19.69 | 3.19 | 36.71 | 20.63 |
> | Pos-HighHES | 35.42 | 27.29 | 17.92 | 18.13 | 11.88 | 19.88 | 4.31 | 35.54 | **21.30** |
> | Pos-Medium-Difficulty | 35.00 | 24.79 | 16.88 | 13.77 | 14.17 | 18.94 | 3.75 | 34.88 | 20.27 |
>
> **Conclusion:**
>
> 1. **Limitations of Difficulty Strategy:** In RFT, Difficulty (27.31%) performs poorly. In RL, Pos-Medium-Difficulty (20.27%) also fails to surpass Full Batch (20.63%). This confirms that screening based solely on difficulty labels is insufficient to guarantee training quality.
> 2. **Consistent Superiority of HES:** Whether in SFT, RFT, or RL, HES consistently beats difficulty baselines (whether Hardest or Medium). This indicates that HES captures a more intrinsic reasoning quality signal—uncertainty and complexity of the thought process—independent of external difficulty labels. HES essentially automatically finds high-value samples relevant to the model's current capabilities.

---

> ### Author Response · Authors · 2025-11-21
> **Reply to Weakness 2**
>
> > **Weakness 2: Generalization verification in non-mathematical domains (Code and STEM).**
>
> We appreciate you pointing out this critical limitation. We strongly agree with your viewpoint: as a framework offering a "Unified" data selection method, validation solely in the mathematics domain is indeed insufficient.
>
> To demonstrate that HES can generalize to other logic-intensive reasoning tasks, we supplemented experiments targeting **Code Generation** and **Scientific Reasoning (STEM)** during the rebuttal period. The experimental results indicate that HES performs exceptionally well in these new domains, even surpassing the improvement magnitude observed in the mathematics domain on certain metrics.
>
> ### 1. Code Domain
> We used the `codeforces-cots` dataset for training, with other experimental settings identical to those in the mathematics domain in the paper. We evaluated model performance on LiveCodeBench, AIME 2025, and GPQA Diamond.
>
> **Table 1: SFT Performance in Code Domain**
>
> | Method | LiveCodeBench_v5 | AIME 2025 | GPQA| Avg |
> | :--- | :---: | :---: | :---: | :---: |
> | Fullset (100%) | 58.76 | 20.00 | 30.08 | 36.28 |
> | Random (20%) | 49.06 | 22.29 | 35.45 | 35.60 |
> | Highest-HES (20%) | 54.59 | 25.00 | 39.02 | **39.54** |
> | Highest-HES (80%) | 61.38 | 23.54 | 33.62 | 39.51 |
> | Lowest-HES (20%) | 35.32 | 12.92 | 29.36 | 25.86 |
>
> **Analysis of Results:**
>
> * **Less is More:** Using only the Top 20% data selected by HES, the average accuracy reaches **39.54%**, which not only significantly outperforms Random-20% (35.60%) but also substantially surpasses training on the full dataset (Fullset: 36.28%).
> * **Data Denoising:** Using the Highest-HES (80%) strategy (i.e., removing the lowest 20% HES data), performance on LiveCodeBench reaches a peak of **61.38%**, superior to the full dataset baseline.
> * **Identification of Low-Quality Data:** The performance of Lowest-HES drops drastically (Avg 25.86%), proving that HES can effectively identify low-quality samples within code data.
>
> ### 2. Scientific Reasoning Domain (STEM)
> We used the STEM subset from the `Llama-Nemotron-Post-Training-Dataset` for training, with other experimental settings identical to those in the mathematics domain in the paper. We conducted evaluations on MMLU-STEM, GPQA Diamond, and HMMT25.
>
> **Table 2: SFT Performance in STEM Domain**
>
> | Method | MMLU-STEM | GPQA| HMMT25 | Avg |
> | :--- | :---: | :---: | :---: | :---: |
> | Fullset (100%) | 85.38 | 47.66 | 0.21 | 44.42 |
> | Random (20%) | 85.82 | 45.86 | 5.63 | 45.77 |
> | Highest-HES (20%) | 88.77 | 50.95 | 8.96 | **49.56** |
> | Highest-HES (80%) | 86.08 | 49.75 | 0.63 | 45.48 |
> | Lowest-HES (20%) | 74.88 | 31.44 | 0.21 | 35.51 |
>
> **Analysis of Results:**
>
> * **Cross-Domain Consistency:** Consistent with the mathematics and code domains, Highest-HES (20%) also achieved the best performance in the STEM domain (Avg **49.56%**), significantly outperforming Fullset (44.42%) and Random (45.77%).
> * **Improvement on GPQA:** On the highly challenging graduate-level scientific QA (GPQA), Highest-HES reached **50.95%**, exceeding the full dataset by 3.29%. This further confirms that HES can select scientific QA samples with high reasoning value.
>
> **Conclusion:** The supplementary experiments forcefully address the query regarding "evaluation limited to the mathematics domain." The results show that HES is effective not only for mathematics but also capable of capturing high-quality thinking patterns (i.e., high thought-density forking points) universal to **Code Logic** and **Scientific Reasoning**. This fully supports the claim of our "Unified Data Selection". We will incorporate these significant cross-domain results into the revised version.

---

> ### Author Response · Authors · 2025-11-21
> **Reply to Weakness 3**
>
> > **Weakness 3: Incomplete Competitive Analysis in RFT and RL**
>
> We greatly appreciate you pointing out the inconsistency in experimental design. We agree that to verify the true effectiveness of HES, the experiments in the RFT and RL phases should include rigorous baseline comparisons similar to the SFT phase, especially on the two critical dimensions of **Length** and **Difficulty**.
>
> To address this, we re-ran the RFT and RL experiments during the rebuttal period, supplementing them with a complete set of strong baseline comparisons. The experimental results show that even against these strong baselines, HES continues to demonstrate significant advantages.
>
> ### 1. Complete Competitive Analysis in RFT
> In the RFT experiment, we added selection strategies based on **Longest Response** and **Medium Difficulty**.
>
> **Table 1: Complete Comparison of RFT@2 Selection Strategy Performance**
>
> | Selection Strategy | AIME 24 | AIME 25 | HMMT 23 | HMMT 24 | HMMT 25 | Oly (Easy) | Oly (Hard) | GPQA | Avg |
> | :--- | :---: | :---: | :---: | :---: | :---: | :---: | :---: | :---: | :---: |
> | **Per-Query** | | | | | | | | | |
> | Random | 46.46 | 34.17 | 31.88 | 28.54 | 21.46 | 35.63 | 4.31 | 40.50 | 30.37 |
> | Highest-HES | 48.33 | 34.58 | 34.38 | 28.13 | 21.67 | 38.06 | 5.56 | 40.30 | **31.38** |
> | Lowest-HES | 43.75 | 34.17 | 30.63 | 25.00 | 18.96 | 33.81 | 4.44 | 39.43 | 28.77 |
> | Longest | 46.04 | 33.33 | 33.75 | 28.13 | 19.58 | 35.94 | 4.81 | 40.56 | 30.27 |
> | **Global Pool** | | | | | | | | | |
> | Random | 42.29 | 31.04 | 30.00 | 26.88 | 18.54 | 30.81 | 4.75 | 38.35 | 27.83 |
> | Highest-HES | 46.46 | 34.58 | 32.50 | 25.42 | 20.83 | 34.00 | 5.31 | 42.30 | **30.18** |
> | Lowest-HES | 19.38 | 14.58 | 10.21 | 11.04 | 7.92 | 10.81 | 2.00 | 31.19 | 13.39 |
> | Medium-Difficulty | 45.00 | 32.71 | 28.13 | 23.54 | 17.71 | 30.13 | 4.31 | 36.93 | 27.31 |
> | Longest | 45.21 | 35.42 | 27.92 | 25.21 | 18.33 | 32.13 | 5.13 | 27.05 | 27.05 |
>
> **Analysis of Results:**
>
> * **HES Outperforms All Strong Baselines:** Whether in Per-Query or Global Pool settings, HES achieved the highest scores.
> * **Limitations of Difficulty and Length Screening:** In the highly challenging Global Pool setting, the performance of **Medium-Difficulty** (27.31%) and **Longest** (27.05%) was even slightly lower than **Random** (27.83%). This indicates that selecting problems purely based on length or difficulty (even with correct answers) does not guarantee optimal training results and may instead introduce noise.
>
> ### 2. Complete Competitive Analysis in RL
> In the RL experiment, we compared the effects of adopting **Longest** and **Medium Difficulty** strategies for positive sample selection.
>
> **Table 2: Complete Comparison of RL Positive Sample Sampling Strategy Performance** (Negative samples all use Random strategy)
>
> | Sampling Strategy | AIME 24 | AIME 25 | HMMT 23 | HMMT 24 | HMMT 25 | Oly (Easy) | Oly (Hard) | GPQA | Avg |
> | :--- | :---: | :---: | :---: | :---: | :---: | :---: | :---: | :---: | :---: |
> | Baseline (Full Batch) | 33.33 | 25.63 | 17.30 | 14.00 | 15.21 | 19.69 | 3.19 | 36.71 | 20.63 |
> | Pos-HighHES | 35.42 | 27.29 | 17.92 | 18.13 | 11.88 | 19.88 | 4.31 | 35.54 | **21.30** |
> | Pos-Rand | 32.30 | 25.21 | 13.54 | 14.11 | 13.96 | 19.18 | 4.25 | 36.49 | 19.88 |
> | Pos-Difficulty | 35.00 | 24.79 | 16.88 | 13.77 | 14.17 | 18.94 | 3.75 | 34.88 | 20.27 |
> | Pos-Longest | 31.04 | 26.25 | 16.46 | 13.95 | 14.79 | 20.00 | 4.06 | 35.32 | 20.23 |
>
> **Analysis of Results:**
>
> * **Only HES Surpasses Full Batch:** Among all sampling strategies, only **Pos-HighHES** (21.30%) successfully surpassed the full data training baseline (Full Batch: 20.63%).
> * **Strong Baselines Failed to Surpass Full Batch:** Although **Pos-Difficulty** (20.27%) and **Pos-Longest** (20.23%) slightly outperformed random sampling (19.88%), they failed to reach the level of Full Batch. This further confirms the uniqueness of HES as a "training-free reward signal"—it can identify samples most valuable for model parameter updates, rather than just difficult or long ones.
>
> ### 3. Consistency Supplement in SFT Phase
> To address concerns about insufficient baselines in the SFT experiment, we also compared **Medium Difficulty**, **Highest Difficulty**, and **Longest** in the SFT phase (see response to Weakness 2 and the revised version for details). The results similarly show that HES significantly outperforms them, maintaining cross-paradigm consistency.
>
> In conclusion, by completing the competitive baselines for SFT, RFT, and RL, we have proven that HES consistently beats length and difficulty heuristic strategies across all training stages. We will incorporate these complete comparison tables into the paper to ensure the rigor and consistency of the experimental design.

---

> ### Author Response · Authors · 2025-11-21
> **Reply to Question 1**
>
> > **Question 1: Experiments on Other Domains (Code & STEM).**
>
> We sincerely appreciate your suggestion regarding generalization in non-mathematical domains. This is crucial for verifying whether HES merits the term "Unified" in the title. Although the original paper focused primarily on mathematical reasoning, the core hypothesis of HES—that "high-entropy forking points reflect critical decisions in the reasoning process"—should logically hold true in other logic-intensive tasks.
>
> To verify this, we supplemented SFT experiments on **Code Generation** and **Scientific Reasoning (STEM)** during the rebuttal period, as detailed in Weakness 2.
>
> **Experimental Settings:**
>
> * **Code Domain:** Training data sourced from `codeforces-cots`; evaluation benchmarks include LiveCodeBench, AIME 2025, and GPQA.
> * **STEM Domain:** Training data sourced from `Llama-Nemotron-Post-Training-Dataset`; evaluation benchmarks include MMLU-STEM, GPQA, and HMMT.
>
> **Experimental Results:**
>
> We compared the performance of the full dataset (Fullset), random sampling (Random), and HES selection strategies. The results are shown in the table below:
>
> **Table 1: Average SFT Performance Comparison in Non-Math Domains (Code & STEM)**
>
> | Method (SFT) | Code Domain (Avg) | STEM Domain (Avg) |
> | :--- | :---: | :---: |
> | Fullset (100%) | 36.28 | 44.42 |
> | Random (20%) | 35.60 | 45.77 |
> | Highest-HES (20%) | **39.54** | **49.56** |
> | Highest-HES (80%) | 39.51 | 45.48 |
> | Lowest-HES (20%) | 25.86 | 35.51 |
>
> **Analysis of Results and Conclusion:**
>
> 1.  **Consistent improvement across domains:** In both Code and STEM domains, using only the Top-20% data selected by HES significantly outperforms the average model performance of training on the Fullset (Code: +3.26%, STEM: +5.14%). This powerfully proves that HES applies not only to mathematics but can also effectively capture high-quality reasoning chains in code logic and scientific reasoning.
> 2.  **Significant "Denoising" Effect:**
>     * Highest-HES (80%) (i.e., removing the lowest 20% HES data) outperforms Fullset in both domains, proving that low-HES data indeed constitutes training noise.
>     * The Lowest-HES (20%) strategy leads to severe performance degradation, confirming that HES can accurately identify low-information or harmful samples in non-math corpora.
>
> In summary, HES demonstrates extremely strong cross-domain generalization capabilities and is a universal data selection metric suitable for broad reasoning tasks. We will incorporate these key results into the paper to support the core claim of "Unified Data Selection."

---

> ### Author Response · Authors · 2025-11-21
> **Reply to Question 2**
>
> > **Question 2: RFT and RL Experiments with Difficulty Baselines.**
>
> We greatly appreciate your profound insight regarding difficulty selection, especially the hypothesis that "Medium Difficulty (Pass rate >0% but <50%) might outperform the hardest samples." This is a classic perspective in curriculum learning.
>
> To verify this, we not only added difficulty baselines in RFT and RL but also specifically subdivided "Medium Difficulty" and "Highest Difficulty" for comparison in the SFT phase. The experimental results indicate that while difficulty is an effective metadata metric, in Long-CoT reasoning tasks, HES, as an Intrinsic Quality Metric, consistently outperforms difficulty indicators based on outcome statistics.
>
> ### 1. SFT Phase: Medium Difficulty vs. Highest Difficulty
>
> To directly address your hypothesis regarding "Medium Difficulty," we compared the performance of different difficulty ranges in the SFT experiment:
>
> **Table 1: Performance Comparison of Difficulty Ranges in SFT**
>
> | Method (SFT) | Avg Accuracy (%) |
> | :--- | :---: |
> | Fullset (100%) | **32.61** |
> | Highest-HES (20%) | 31.14 |
> | Highest-Difficulty (20%) | 29.88 |
> | Medium-Difficulty (20%) | 23.29 |
>
> Surprisingly, in this task, the performance of Medium-Difficulty (23.29%) was actually lower than Highest-Difficulty (29.88%). This might be because, for stimulating reasoning capabilities, medium-difficulty samples often possess shorter paths or more fixed patterns, lacking sufficient "high-entropy forking points" to train the model's generalization ability. In contrast, HES still holds the lead.
>
> ### 2. RFT Phase: Difficulty Baseline Comparison
>
> In the Global Pool setting of RFT, we compared HES with the difficulty baseline:
>
> **Table 2: RFT@2 Performance Comparison (Global Pool)**
>
> | Selection Strategy (RFT) | Avg Accuracy (%) |
> | :--- | :---: |
> | Highest-HES | **30.18** |
> | Medium-Difficulty | 27.31 |
>
> * **Analysis:** Screening based on difficulty (27.31%) performed relatively poorly in this scenario. This further proves that relying solely on pass rates to select data cannot guarantee the selection of high-quality reasoning trajectories. HES (30.18%) achieved significant improvement by directly evaluating the "thought density" of the reasoning process.
>
> ### 3. RL Phase: Difficulty Baseline Comparison
>
> In RL, we compared the effects of using difficulty strategies for Positive Sampling:
>
> **Table 3: RL Positive Sampling Strategy Comparison**
>
> | Sampling Strategy (RL) | Avg Accuracy (%) | vs. Full Batch |
> | :--- | :---: | :---: |
> | Full Batch (Baseline) | 20.63 | - |
> | Pos-HighHES | **21.30** | **+0.67** |
> | Pos-Difficulty | 20.27 | -0.36 |
>
> * **Analysis:** Pos-Difficulty (20.27%) also failed to surpass the Full Batch baseline (20.63%). Only Pos-HighHES (21.30%) successfully outperformed the full dataset training.
>
> **Conclusion:** The experimental results consistently show that whether selecting "Medium Difficulty" or "Highest Difficulty" in SFT, or using difficulty baselines in RFT/RL, their effects cannot match HES. HES is able to automatically capture those samples with the most "learning value" for the model's current capabilities (i.e., samples containing critical decision points), without relying on external difficulty statistical information.

---

### Official Review · Reviewer_vsxf · 2025-11-05

**Soundness:** 3
**Presentation:** 3
**Contribution:** 2
**Rating:** 4
**Confidence:** 4

**Summary:**

The paper proposes High-Entropy Sum (HES), a training-free metric for data/trajectory selection in LLM reasoning. HES sums only the top-p% highest-entropy tokens in a response to capture “forking points” where the model is uncertain. The authors use HES to (i) rank SFT data, (ii) pick candidates for RFT, and (iii) design an asymmetric RL sampling scheme (select high-HES positives, random negatives). On math-reasoning datasets (Open-Math-Reasoning, Open-R1-220k) and math/STEM benchmarks (AIME24/25, HMMT23/24/25, OlymMATH, GPQA), they report that training on top-20% HES data matches or exceeds full-dataset SFT and that pruning the bottom-20% improves over full-dataset; HES also beats AvgEntropy, AvgEntropy over high-entropy tokens, and total entropy baselines. RL experiments show the “Pos-High, Neg-Rand” recipe outperforming full-batch despite using half the rollouts.

**Strengths:**

1. HES is easy to compute from token distributions and is used consistently across SFT/RFT/RL without training additional selectors or reward models. This practical unification is appealing for data-centric pipelines.

2. SFT/RFT ablations are thorough (top/bottom slices, per-query vs global pools, multiple k) and show sizeable gaps between High-HES vs Random/Low-HES; tables make the effect sizes easy to read.

3. The asymmetric sampling result—keep diverse negatives but focus on high-HES positives—provides a practical recipe that beats full-batch with fewer trajectories, which many practitioners would value.

**Weaknesses:**

1. All datasets and most benchmarks are math; models are Qwen3 variants and DeepSeek-R1 distilled families. It is unclear whether HES transfers to non-math reasoning (code, science QA, multi-modal) or to very different model families.

2. While there is a small sensitivity plot, the theoretical or data-driven rationale for the default percentile and its stability across lengths/models/tasks is limited; AvgHE vs HES comparisons help, but a more systematic hyperparameter study across domains is missing.

3. The pipeline often first filters to correct rollouts, which blurs whether HES measures quality or just difficulty/length. Tighter decontamination and simple causal checks (e.g., control for length/difficulty; test on non-math tasks) are needed to show HES truly beats global metrics.

**Questions:**

-- How does HES perform on non-math corpora (e.g., code generation with execution checks, GSM8K-style verbal math vs olympiad math, commonsense multi-step QA, open-ended planning)?

-- Does HES still help when the base model already produces short CoT or when responses are heavily compressed?

-- How sensitive are results to sampling temperature/max length, which change token-entropy distributions? Any normalization by length or by per-token calibration?

---

> ### Author Response · Authors · 2025-11-21
> **Reply to Weakness 1 --- Part 1**
>
> Dear reviewer,
>
> Thank you very much for your careful review of our work. We also sincerely appreciate the valuable points you have raised, which will be of great help in improving our paper. We will address each of your points in detail as follows.
>
> > **Weakness 1: Generalization regarding non-math domains and different model architectures.**
>
> We sincerely appreciate you pointing out the question regarding the generalization ability of HES on non-math fields and different model families. This is a crucial issue. Although the original paper primarily focused on mathematical reasoning tasks (as they represent the current primary application scenario for Long-CoT), to demonstrate the validity of HES as a general-purpose metric, we have supplemented experiments for **Code** and **Science (STEM)** domains, as well as experiments for the **Llama model family** during the rebuttal period. Aside from the data sources, the training hyperparameters (such as epoch, learning rate, batch size) for all new experiments remain consistent with the mathematical settings in the paper.
>
> The newly added experimental results strongly support the generalization capability of HES:
>
> ### 1. Domain Generalization: Significant Improvement on Code and STEM Tasks
>
> To verify the performance of HES in non-mathematical reasoning tasks, we introduced the following two datasets for SFT experiments:
>
> * **Code Dataset:** Sourced from the `codeforces-cots` dataset.
> * **STEM Dataset:** Sourced from the `Llama-Nemotron-Post-Training-Dataset`.
>
> The experimental setup remains consistent with the main experiments in the paper, comparing the Fullset, Random selection, and HES selection strategies. The experimental results are shown in the tables below:
>
> **Table 1: SFT Performance in Code Domain**
>
> | Method | LiveCodeBench_v5 | AIME 2025 | GPQA | Avg |
> | :--- | :---: | :---: | :---: | :---: |
> | Fullset (100%) | 58.76 | 20.00 | 30.08 | 36.28 |
> | Random (20%) | 49.06 | 22.29 | 35.45 | 35.60 |
> | Highest-HES (20%) | 54.59 | 25.00 | 39.02 | **39.54** |
> | Highest-HES (80%) | 61.38 | 23.54 | 33.62 | 39.51 |
> | Lowest-HES (20%) | 35.32 | 12.92 | 29.36 | 25.86 |
>
> **Table 2: SFT Performance in STEM Domain**
>
> | Method | MMLU-STEM | GPQA | HMMT25 | Avg |
> | :--- | :---: | :---: | :---: | :---: |
> | Fullset (100%) | 85.38 | 47.66 | 0.21 | 44.42 |
> | Random (20%) | 85.82 | 45.86 | 5.63 | 45.77 |
> | Highest-HES (20%) | 88.77 | 50.95 | 8.96 | **49.56** |
> | Highest-HES (80%) | 86.08 | 49.75 | 0.63 | 45.48 |
> | Lowest-HES (20%) | 74.88 | 31.44 | 0.21 | 35.51 |
>
> **Analysis of Results:**
>
> * **Superior to Random and Fullset:** In Code and STEM domains, the Highest-HES (20%) strategy significantly outperforms Random (20%) on the vast majority of benchmarks. In STEM tasks, training with only 20% of high-HES data even comprehensively surpasses the performance of training on the full dataset (Fullset). In Code tasks, using Highest-HES (80%) for denoising improves the performance on LiveCodeBench from 58.76% (Fullset) to 61.38%.
> * **Identification of Low-Quality Data:** Whether in Code or STEM, the performance of Lowest-HES (20%) is substantially lower than the Random baseline, confirming that HES can effectively identify low-information or harmful samples across domains.

---

> ### Author Response · Authors · 2025-11-21
> **Reply to Weakness 1 --- Part 2**
>
> ### 2. Model Architecture Generalization: Validation on the Llama-3.1-8B Base
>
> Addressing your concern regarding "models primarily based on Qwen variants," we would first like to clarify: the choice of Qwen was made because the current mainstream Long-CoT models (such as DeepSeek-R1 and its distilled series) are primarily built upon the Qwen base, representing the current SOTA level.
>
> However, to demonstrate that HES does not rely on a specific architecture, we supplemented RFT experiments on a **DeepSeek Distill model based on Llama-3.1-8B**.
>
> **Table 3: RFT@4 Performance Based on Llama-3.1-8B**
>
> | Selection Strategy | AIME 24 | AIME 25 | HMMT 23 | HMMT 24 | HMMT 25 | Oly (Easy) | Oly (Hard) | GPQA | Avg |
> | :--- | :---: | :---: | :---: | :---: | :---: | :---: | :---: | :---: | :---: |
> | **Per-Query** | | | | | | | | | |
> | Random | 17.08 | 22.50 | 10.63 | 12.71 | 9.17 | 12.38 | 2.44 | 27.81 | 14.34 |
> | Highest-HES | 18.54 | 19.38 | 14.79 | 10.83 | 11.88 | 14.31 | 2.88 | 28.13 | **15.09** |
> | Lowest-HES | 14.58 | 18.13 | 12.08 | 10.21 | 9.38 | 12.06 | 2.06 | 27.30 | 13.23 |
> | **Global Pool** | | | | | | | | | |
> | Random | 14.17 | 19.58 | 11.04 | 11.46 | 10.56 | 12.81 | 3.35 | 27.05 | 13.75 |
> | Highest-HES | 17.92 | 20.63 | 14.38 | 11.25 | 12.29 | 15.19 | 3.13 | 25.09 | **17.92** |
> | Lowest-HES | 1.46 | 10.00 | 5.21 | 5.21 | 2.92 | 3.56 | 1.31 | 27.43 | 7.14 |
>
> **Analysis of Results:**
>
> * **Cross-Architecture Robustness:** On the Llama base, the Highest-HES strategy consistently outperforms the Random baseline in both local (Per-Query) and global (Global Pool) selection.
> * **Extreme Discrimination:** Particularly under the most difficult **Global Pool** setting (which involves screening not only among different responses to the same question but also among problems of varying difficulty), the Lowest-HES strategy leads to a catastrophic drop in model performance to **7.14%** (nearly half compared to the Random baseline of 13.75%). This powerfully demonstrates that HES does not rely on the characteristics of the Qwen model, but rather captures the intrinsic uncertainty features of the reasoning process.
>
> In summary, the above supplementary experiments clearly indicate that HES is not a specialized metric optimized solely for math tasks or the Qwen architecture. Whether processing code logic, scientific reasoning, or transferring to the Llama base, HES demonstrates powerful robustness and discriminative ability as a **unified data selection metric**. We will include these significant results in the final version of the paper.

---

> ### Author Response · Authors · 2025-11-21
> **Reply to Weakness 2**
>
> > **Weakness 2: Hyperparameter stability and length-controlled variable analysis.**
>
> We appreciate your profound question regarding hyperparameter settings and their relationship with sample length. To demonstrate that HES is not merely a proxy metric for length and to validate the robustness of its default hyperparameters, we conducted two additional sets of experiments.
>
> ### 1. Length-Controlled Variable Analysis
> You raised the concern that HES might simply tend to select longer reasoning paths. To decouple the relationship between length and HES, we designed a rigorous stratified control experiment on the OpenMathReasoning dataset:
>
> 1. We divided the dataset into three groups based on token length: Short, Medium, and Long.
> 2. Within each group, we selected the top 20% (Highest-HES) and bottom 20% (Lowest-HES) based on HES scores and performed SFT training on Qwen3-8B.
>
> The experimental results are shown in the table below:
>
> **Table 1: Performance Comparison of HES Under Different Length Groups**
> (Comparing Highest-HES vs Lowest-HES within the same length interval)
>
> | Length Group | Selection | AIME 24 | AIME 25 | HMMT 23 | HMMT 24 | HMMT 25 | Oly (Easy) | Oly (Hard) | GPQA | Avg |
> | :--- | :--- | :--- | :--- | :--- | :--- | :--- | :--- | :--- | :--- | :--- |
> | **Long** | Highest-HES | 40.42 | 31.88 | 26.67 | 22.71 | 18.75 | 31.38 | 4.50 | 48.14 | **28.06** |
> | | Lowest-HES | 35.00 | 30.21 | 31.88 | 22.92 | 20.83 | 28.88 | 4.63 | 47.98 | 27.79 |
> | **Medium** | Highest-HES | 40.63 | 29.17 | 25.63 | 23.75 | 19.17 | 28.44 | 3.38 | 46.34 | **27.06** |
> | | Lowest-HES | 26.67 | 23.96 | 17.08 | 13.54 | 16.04 | 16.25 | 2.88 | 40.75 | 19.65 |
> | **Short** | Highest-HES | 28.13 | 20.42 | 17.08 | 12.08 | 12.50 | 16.75 | 2.00 | 41.35 | **18.79** |
> | | Lowest-HES | 11.88 | 10.00 | 1.67 | 6.46 | 1.67 | 5.63 | 1.19 | 36.08 | 9.32 |
>
> **Analysis of Results:**
>
> * **HES is effective across all length groups:** Whether in the Long, Medium, or Short group, Highest-HES consistently outperforms Lowest-HES. This directly proves that HES captures the thought density of critical forking points in the reasoning process, rather than just length.
> * **Extremely high discrimination in Short/Medium samples:** Notably, in the Short group, the performance of Highest-HES (18.79%) is nearly **double** that of Lowest-HES (9.32%). This indicates that even in shorter reasoning processes, whether high-information "thought jumps" or "critical decisions" are included is vital for model training.
> * **Robustness in long samples:** In the Long group, the gap between Highest-HES and Lowest-HES is relatively smaller. This may be because long samples inherently contain more reasoning steps, meaning even relatively low-HES long samples contain some effective information, yet Highest-HES still maintains a slight advantage.
>
> ### 2. Cross-Domain Hyperparameter Stability Validation
> Addressing your point regarding the "lack of systematic hyperparameter study across domains," we replicated the sensitivity analysis experiment from Figure 3 of the paper in the **STEM and Code domain** (see Figure 4 in the revised paper).
>
> We tested different High-Entropy Token Ratios (e.g., 0.005, 0.05, 0.5, 1). The experimental results are highly consistent with the conclusions from the mathematics domain:
>
> * In other tasks, optimal performance generally appears around the **Top-0.5% (0.005)** ratio.
> * This further validates our theoretical hypothesis: Critical forking points (Forking Points) in the reasoning process belong to the extremely sparse long tail of the distribution. This pattern applies not only to mathematical reasoning but also widely exists in scientific reasoning and code generation.
>
> **Conclusion:** Through length-stratified experiments and cross-domain hyperparameter validation, we have confirmed that HES is an independent metric reflecting reasoning quality, independent of sample length, and its core hyperparameter (0.5%) possesses good cross-domain robustness.

---

> ### Author Response · Authors · 2025-11-21
> **Reply to Weakness 3 --- Part 1**
>
> > **Weakness 3: HES is not a proxy metric for length or difficulty.**
>
> You raised a very pertinent question: Does selecting correct answers before calculating HES cause HES to merely become a proxy metric for length or difficulty? To clarify the causal relationship between HES and these two factors, in addition to the previously mentioned SFT length stratification experiment, we introduced explicit "Length" and "Difficulty" baselines for comparison in the RFT and RL phases.
>
> The experimental results consistently show that screening based solely on length or difficulty cannot replicate the performance improvement of HES, proving that HES captures a unique signal of reasoning quality.
>
> ### 1. Comparative Experiment in RFT Phase: HES vs. Length/Difficulty
>
> In the RFT experiment, we compared selection strategies based on Highest HES, Longest Token Count, and Difficulty (selecting medium difficulty as suggested by Reviewer rK1A).
>
> **Table 1: Comprehensive Comparison of RFT@2 Selection Strategy Performance**
>
> | Selection Strategy | AIME 24 | AIME 25 | HMMT 23 | HMMT 24 | HMMT 25 | Oly (Easy) | Oly (Hard) | GPQA | Avg |
> | :--- | :---: | :---: | :---: | :---: | :---: | :---: | :---: | :---: | :---: |
> | **Per-Query** | | | | | | | | | |
> | Random | 46.46 | 34.17 | 31.88 | 28.54 | 21.46 | 35.63 | 4.31 | 40.50 | 30.37 |
> | Highest-HES | 48.33 | 34.58 | 34.38 | 28.13 | 21.67 | 38.06 | 5.56 | 40.30 | **31.38** |
> | Lowest-HES | 43.75 | 34.17 | 30.63 | 25.00 | 18.96 | 33.81 | 4.44 | 39.43 | 28.77 |
> | Length | 46.04 | 33.33 | 33.75 | 28.13 | 19.58 | 35.94 | 4.81 | 40.56 | 30.27 |
> | **Global Pool** | | | | | | | | | |
> | Random | 42.29 | 31.04 | 30.00 | 26.88 | 18.54 | 30.81 | 4.75 | 38.35 | 27.83 |
> | Highest-HES | 46.46 | 34.58 | 32.50 | 25.42 | 20.83 | 34.00 | 5.31 | 42.30 | **30.18** |
> | Lowest-HES | 19.38 | 14.58 | 10.21 | 11.04 | 7.92 | 10.81 | 2.00 | 31.19 | 13.39 |
> | Difficulty | 45.00 | 32.71 | 28.13 | 23.54 | 17.71 | 30.13 | 4.31 | 36.93 | 27.31 |
> | Length | 45.21 | 35.42 | 27.92 | 25.21 | 18.33 | 32.13 | 5.13 | 27.05 | 27.05 |
>
> **Analysis of Results:**
>
> * **Failure of Length and Difficulty Screening in Global Pool:** Under the most challenging Global Pool setting, directly screening for the longest response (Avg 27.05%) or the most difficult problem (Avg 27.31%) yielded performance even lower than random selection (Avg 27.83%). This indicates that within a pool of correct answers, simply pursuing long reasoning chains or difficult problems does not translate into improved model capability, and may even have a negative impact by introducing ineffective verbosity or overly difficult outliers.
> * **Uniqueness and Effectiveness of HES:** In contrast, Highest-HES achieved an average accuracy of **30.18%**, significantly surpassing the random baseline and the length/difficulty baselines. This proves that the high-entropy samples selected by HES are not merely "long" or "difficult," but contain "high thought density" features that are more valuable for model training.

---

> ### Author Response · Authors · 2025-11-21
> **Reply to Weakness 3 --- Part 2**
>
> ### 2. Comparative Experiment in RL Phase: Positive Sample Sampling Strategy
>
> In the RL experiment (GRPO), we maintained the random sampling strategy for negative samples (Neg-Rand) and added strategies for selecting positive samples (Positive) based on length and difficulty.
>
> **Table 2: Comprehensive Comparison of RL Positive Sample Sampling Strategy Performance**
>
> | Sampling Strategy | AIME 24 | AIME 25 | HMMT 23 | HMMT 24 | HMMT 25 | Oly (Easy) | Oly (Hard) | GPQA | Avg |
> | :--- | :---: | :---: | :---: | :---: | :---: | :---: | :---: | :---: | :---: |
> | Baseline (Full Batch) | 33.33 | 25.63 | 17.30 | 14.00 | 15.21 | 19.69 | 3.19 | 36.71 | 20.63 |
> | Pos-HighHES, Neg-Rand | 35.42 | 27.29 | 17.92 | 18.13 | 11.88 | 19.88 | 4.31 | 35.54 | **21.30** |
> | Pos-Longest, Neg-Rand | 31.04 | 26.25 | 16.46 | 13.95 | 14.79 | 20.00 | 4.06 | 35.32 | 20.23 |
> | Pos-Difficulty, Neg-Rand | 35.00 | 24.79 | 16.88 | 13.77 | 14.17 | 18.94 | 3.75 | 34.88 | 20.27 |
>
> **Analysis of Results:**
>
> * **HES is the Only Strategy Surpassing Full Batch:** Among all sampling strategies, only Pos-HighHES, Neg-Rand (Avg **21.30%**) successfully surpassed the full dataset training baseline (Baseline: 20.63%).
> * **Limitations of Length and Difficulty Strategies:** Selecting positive samples based on length (Pos-Longest: 20.23%) or moderate difficulty (selecting medium difficulty as suggested by Reviewer rK1A, Pos-Difficulty: 20.27%) failed to reach the level of Full Batch. This result holds not only for the overall average but also across most individual items, where HES demonstrated a clear advantage over length and difficulty strategies.
>
> **Conclusion:** Combining the strictly controlled variable experiments across the three phases of SFT, RFT, and RL, we can confirm: HES is not a proxy metric for length or difficulty. By locating "critical forking points" in the reasoning process, it provides a quality metric orthogonal to length and difficulty, which is why it consistently outperforms other baselines and serves as an efficient and robust reward signal.

---

> ### Author Response · Authors · 2025-11-21
> **Reply to Question 1**
>
> > **Question 1: Performance on Non-Math Corpora (Code & STEM).**
>
> We sincerely appreciate your question regarding generalization in non-math domains. Although the original paper primarily focused on mathematical reasoning, the core hypothesis of HES—that "high-entropy forking points reflect critical decisions in the reasoning process"—should logically hold true in other logic-intensive tasks. To verify this, we conducted supplementary SFT experiments on Code Generation and Scientific Reasoning (STEM) during the rebuttal period.
>
> We compared the performance of the full dataset (Fullset), random sampling (Random), and HES selection strategies. The results are shown in the table below:
>
> **Table 1: Average SFT Performance Comparison in Non-Math Domains (Code & STEM)**
>
> | Method (SFT) | Code Domain (Avg) | STEM Domain (Avg) |
> | :--- | :---: | :---: |
> | Fullset (100%) | 36.28 | 44.42 |
> | Random (20%) | 35.60 | 45.77 |
> | Highest-HES (20%) | **39.54** | **49.56** |
> | Lowest-HES (20%) | 25.86 | 35.51 |
>
> **Conclusion:**
>
> 1. **Consistent Improvement Across Domains:** In both Code and STEM domains, using only the Top-20% data selected by HES significantly outperforms Random-20% and even substantially surpasses the performance of training on the full dataset (Fullset) (Code: +3.26%, STEM: +5.14%). This indicates that HES can effectively capture high-quality reasoning chains in code logic and scientific reasoning.
> 2. **Strong Negative Sample Identification:** The Lowest-HES strategy resulted in severe performance degradation (dropping to 25.86% in the Code domain and 35.51% in the STEM domain), confirming that HES can accurately identify low-information or simple samples in non-math corpora.
>
> In summary, HES is not limited to the mathematics domain but serves as a universal data selection metric applicable to broad reasoning tasks. We will incorporate these results into the paper to enhance the universality of our conclusions.

---

> ### Author Response · Authors · 2025-11-21
> **Reply to Question 2**
>
> > **Question 2: Effectiveness on Short or Compressed CoT.**
>
> This is indeed a very insightful question. To verify whether HES relies solely on Long CoT or if it remains effective for short/compressed responses, we conducted a rigorous length-stratified control experiment on the Open-Math-Reasoning dataset.
>
> We divided the dataset into three groups based on token length: Short, Medium, and Long. Within each group, we selected the top 20% (Highest-HES) and bottom 20% (Lowest-HES) for SFT training.
>
> **Table 1: Performance Comparison of HES Under Different Length Groups**
>
> | Length Group | Highest-HES (20%) | Lowest-HES (20%) | Performance Gap |
> | :--- | :---: | :---: | :---: |
> | Long | 28.06 | 27.79 | +0.27 |
> | Medium | 27.06 | 19.65 | +7.41 |
> | Short | 18.79 | 9.32 | **+9.47** |
>
> **Conclusion:**
>
> 1. **Extremely effective on Short CoT:** The experimental results show that HES actually exhibits the greatest discrimination in the Short group. The performance of Highest-HES (18.79%) is nearly double that of Lowest-HES (9.32%). This indicates that even in shorter reasoning processes, HES is still able to effectively distinguish between "short but concise (high thought density)" and "short and simple (low information)" samples.
> 2. **HES captures thought density rather than length:** If HES were only effective for Long CoT, we should see negligible gaps in the short group. On the contrary, the data proves that when responses are compressed or inherently short, identifying samples containing critical "forking points" (high-entropy tokens) is even more critical for maintaining model reasoning capabilities.
>
> Therefore, HES is not only applicable to Long CoT but remains a powerful quality screening metric when models generate short responses or compressed reasoning paths.

---

> ### Author Response · Authors · 2025-11-21
> **Reply to Question 3 --- Part 1**
>
> > **Question 3: Sensitivity to Sampling Temperature, Length, and Model Scale.**
>
> We appreciate your insightful questions regarding metric sensitivity and normalization mechanisms. Our method was designed with these issues in mind, implementing adaptive normalization for length and distribution shifts through the relative threshold mechanism ($HES_{relative}$).
>
> ### 1. Normalization via Relative Threshold
> Addressing "length variation" and "normalization," the $HES_{relative}$ proposed in Section 3.1 adopts a percentile-based truncation strategy (summing entropy of the top $p\%$ tokens) rather than a fixed number or absolute threshold.
>
> * **Adaptive Mechanism:** This design essentially serves as an implicit Length Normalization. Regardless of the CoT length, HES always focuses on the relatively most "critical" small portion of forking points (Top 0.5%) within that sequence.
> * **Empirical Support:** The **Length Bucket Experiment** presented in our response to Question 2 provides direct evidence. Experimental results show that HES significantly outperforms Random and Bottom baselines across Short, Medium, and Long length groups. This proves that the effectiveness of HES does not depend on sequence length, as its normalization mechanism successfully strips away bias introduced by length.
>
> ### 2. Robustness to Entropy Distribution Shifts
> Regarding changes in Token entropy distribution caused by sampling temperature:
>
> * **Stability of Relative Ranking:** Although changes in temperature scale the predicted probability distribution, thereby altering the absolute values of entropy, HES data selection operates based on relative ranking among samples (Ranking-based Selection, e.g., selecting the Top 20% of samples). As long as the relative uncertainty relationship between high-quality and low-quality reasoning paths remains consistent, HES is inherently robust to global shifts in absolute entropy values.
> * **Comparative Validation:** The results in the main paper showing $HES_{relative}$ outperforming $HES_{absolute}$ also corroborate this. Using an absolute threshold is indeed susceptible to distribution shifts, whereas $HES_{relative}$ effectively resolves this issue by adapting to the distribution characteristics of each sample itself.

---

> ### Author Response · Authors · 2025-11-21
> **Reply to Question 3 --- Part 2**
>
> ### 3. Robustness to Model Scale
> You raised concerns about changes in Token entropy distribution. To test whether HES relies on the probability distribution of specific large models, we conducted cross-model screening experiments on the Open-Math-Reasoning and Open-R1-220k datasets.
>
> We attempted to use Qwen3-0.6B and Qwen3-1.7B, which are significantly smaller than the training model (8B), to calculate HES and select the Top 20% of data, and then used this selected data to fine-tune the 8B model. The results are as follows:
>
> **Table 1: Cross-Model Screening SFT Performance Comparison on Open-Math-Reasoning**
>
> | Method | AIME 24 | AIME 25 | HMMT 23 | HMMT 24 | HMMT 25 | Oly (Easy) | Oly (Hard) | GPQA | Avg |
> | :--- | :---: | :---: | :---: | :---: | :---: | :---: | :---: | :---: | :---: |
> | Fullset (100%) | 50.83 | 34.17 | 35.21 | 28.13 | 24.58 | 42.94 | 6.94 | 38.04 | 32.61 |
> | Random (20%) | 39.79 | 27.92 | 24.79 | 20.21 | 20.42 | 30.00 | 3.88 | 40.09 | 25.89 |
> | Highest-HES (Default) | 47.29 | 36.04 | 33.13 | 25.63 | 22.08 | 36.31 | 5.56 | 43.06 | **31.14** |
> | Lowest-HES (Default) | 18.54 | 18.96 | 11.88 | 11.04 | 7.92 | 11.00 | 2.94 | 36.90 | 14.90 |
> | Highest-HES (via 0.6B) | 49.17 | 35.42 | 32.50 | 24.17 | 22.29 | 41.50 | 6.38 | 45.55 | **32.12** |
> | Highest-HES (via 1.7B) | 47.08 | 34.79 | 31.04 | 26.04 | 21.46 | 40.06 | 4.56 | 45.20 | **31.28** |
>
> **Table 2: Cross-Model Screening SFT Performance Comparison on Open-R1-220k**
>
> | Method | AIME 24 | AIME 25 | HMMT 23 | HMMT 24 | HMMT 25 | Oly (Easy) | Oly (Hard) | GPQA | Avg |
> | :--- | :---: | :---: | :---: | :---: | :---: | :---: | :---: | :---: | :---: |
> | Fullset (100%) | 48.96 | 32.50 | 31.67 | 25.83 | 22.50 | 38.38 | 5.81 | 37.00 | 30.33 |
> | Random (20%) | 30.00 | 30.21 | 24.17 | 19.79 | 17.50 | 25.00 | 5.56 | 41.73 | 24.25 |
> | Highest-HES (Default) | 48.54 | 32.92 | 30.63 | 23.33 | 22.29 | 37.94 | 6.56 | 49.78 | **31.50** |
> | Bottom-HES (Default) | 15.42 | 12.29 | 6.04 | 8.13 | 6.04 | 5.63 | 2.13 | 35.54 | 11.40 |
> | Highest-HES (via 0.6B) | 48.75 | 35.42 | 28.75 | 24.38 | 20.00 | 39.94 | 6.25 | 48.26 | **31.47** |
> | Highest-HES (via 1.7B) | 48.13 | 35.42 | 31.46 | 26.46 | 20.00 | 37.31 | 6.81 | 49.08 | **31.83** |
>
> **Conclusion:**
>
> 1. **Extremely High Consistency and Stability:** Surprisingly, using tiny models with only 0.6B or 1.7B parameters to calculate HES and filter data resulted in final training performance (Avg: 32.12% / 31.47%) that is almost consistent with, and on some metrics even slightly better than, filtering using the 8B model itself (Avg: 31.14% / 31.50%).
> 2. **HES Reflects Intrinsic Data Properties:** This result strongly proves that HES is insensitive to specific numerical values of Token entropy distributions. Even if the probability distribution of a small model is not perfect, it can still accurately identify "high uncertainty forking points" in the reasoning path. This demonstrates that HES captures the logical complexity and learning value intrinsic to the data, rather than artifacts of specific model parameters.
>
> In summary, through the relative threshold design and ranking-based mechanism, HES demonstrates extremely strong robustness. It is not only insensitive to length and temperature but also provides stable and high-quality data selection signals even across massive model scale differences (from 0.6B to 8B).

---

### Author Response · Authors · 2025-11-30
**Summary of Rebuttal and Looking forward to discussing with you**

Dear Area Chair,

Thank you for your time and effort in this challenging review cycle. To facilitate your review, we provide this summary to clarify our rebuttal status. We have achieved a confirmed consensus with Reviewer s4CB (**Score raised 2 → 6**) and conducted comprehensive experiments to fully resolve the concerns of the remaining reviewers who have not yet replied. We are eager to discuss these findings directly with you and clarify any remaining points to support your final decision.

### **1. Confirmed Consensus & Addressed Concerns**

* **Reviewer s4CB (Score 2 → 6):**
    * **Concern:** Questioned whether HES is merely a proxy for length; raised concerns about the computational overhead of calculating entropy; questioned whether small models can screen for large ones.
    * **Resolution:** We provided **length-controlled stratified experiments** demonstrating that HES discriminates data quality within fixed length groups, proving it operates independently of length and remains effective even on **short samples**. We also introduced **Length baselines** in RFT/RL, where HES consistently outperformed length-based strategies. Furthermore, we clarified that **Online RFT/RL incurs zero extra inference cost** (requiring only negligible storage for storage), while **Offline SFT** can be efficiently handled by a **0.6B proxy model**. The reviewer explicitly acknowledged these results as persuasive and raised his/her score.

* **Reviewer vsxf (Score: 4):**
    * **Concern:** Questioned generalization to non-math reasoning; requested controlled experiments for length/difficulty; noted a lack of systematic hyperparameter study across domains.
    * **Resolution:** We added **Code and STEM** experiments, where **Highest-20% HES** outperformed the **Full-Dataset** baseline (Code: +3.26%, STEM: +5.14%). We also implemented **length-controlled stratified experiments** and added **Length/Difficulty baselines** in RFT/RL to decouple HES from these factors, alongside a **sensitivity analysis** in the STEM and Code domain validating our default settings.

* **Reviewer rK1A (Score: 4):**
    * **Concern:** Suggested "Medium Difficulty" baselines; noted the lack of comprehensive competitive analyses (Length/Difficulty) in RFT and RL; questioned the limitation to math datasets.
    * **Resolution:** We implemented **Medium Difficulty** and **Length** baselines across SFT, RFT, and RL. Results show HES consistently outperforms both Medium Difficulty and Length-based selection. We also added **Code and STEM** experiments to prove cross-domain generalization.

* **Reviewer 2B5q (Score: 4):**
    * **Concern:** Questioned the link between Entropy and Data Quality; noted model inconsistency across SFT (Base) vs. RFT/RL (Instruct); questioned generalization to non-math domains.
    * **Resolution:** We clarified that HES captures **intrinsic reasoning complexity**. This is validated by our **small-model transfer experiment**, showing that HES signals are consistent across model scales (0.6B, 1.7B and 8B) and thus reflect objective data quality. We aligned our evaluation by adding **R1-7B SFT** experiments and verified generalization via **Code/STEM** tasks. We also validated model agnosticism using the **Llama architecture**.

### **2. Summary of New Experiments (Included in Revision)**

We mainly conducted **these new experiments** to address all weaknesses raised:

* **Generalization (Code & STEM):** On *Codeforces* and *Llama-Nemotron-STEM* datasets, **Highest-20% HES** data significantly outperformed the **Full-Dataset** baseline (Code: +3.26%, STEM: +5.14%), proving HES is a unified metric beyond math.
* **Robustness vs. Length/Difficulty:** In RFT and RL, we added strong baselines (**Longest Response**, **Medium Difficulty**). HES consistently outperformed them.
* **Mechanism Verification (Length-Control):** Stratified experiments confirmed HES distinguishes quality within length groups. Crucially, on **short samples**, High-HES performance was nearly double that of Low-HES, disproving the "length proxy" hypothesis.
* **Computational Efficiency:** We demonstrated that a **0.6B proxy model** can screen data for an **8B model** with comparable performance (Avg 32.12% vs. 31.14%), solving offline SFT computation concerns. In online RL/RFT, HES incurs **zero extra inference cost** by only small storage overhead.
* **Model Agnosticism:** We validated HES on the **Llama-3.1-8B** architecture (RFT), proving effectiveness is not limited to the Qwen family.
* **Model Consistency:** We replicated SFT experiments using **R1-7B**, the same model series used for our RFT/RL experiments. The results confirm that HES is highly effective (**Highest-20% HES**: 34.61% vs. **Fullset**: 30.22%), aligning our evaluation pipeline.

We believe these results robustly support HES as a **Unified, Efficient, and Effective** data selection framework. Thank you again for your dedication to the review process.

Best regards,

Authors

---

### Meta-Review · Area_Chair_ECzy · 2025-12-27

**Summary:**

This paper introduces a training-free metric for selecting SFT/RFT data. The reviewers primarily raise concerns about the scale of the original experiments and their generalization capabilities. For example, the original paper covers only experiments in mathematics. In the rebuttal, the authors add experimental results on coding and reasoning datasets to validate the effectiveness of the proposed metric, but do not provide the performance of other baseline methods. Since we are to make decisions primarily based on the submitted manuscript, I tend to agree with the reviewers' recommendation to reject this paper.

**Reviewer Concerns:**

The reviewers primarily raise concerns about the scale of the original experiments and their generalization capabilities.

**Reviewer Scores:**

I do not see clear evidence that the reviewers might change their scores.

---

### Decision · Program_Chairs · 2026-01-26

Reject